# Suppression of Vps13 adaptor protein mutants reveals a central role for PI4P in regulating prospore membrane extension

**Tsuyoshi S. Nakamura**[1,2], **Yasuyuki Suda**[3,4], **Kenji Muneshige**[1], **Yuji Fujieda**[1], **Yuuya Okumura**[1], **Ichiro Inoue**[1], **Takayuki Tanaka**[1], **Tetsuo Takahashi**[5], **Hideki Nakanishi**[6], **Xiao-Dong Gao**[6], **Yasushi Okada**[7,8,9], **Aaron M. Neiman**[10], **Hiroyuki Tachikawa**[1,11]*

**1** Department of Applied Biological Chemistry, Graduate School of Agricultural and Life Sciences, The University of Tokyo, Tokyo, Japan, **2** Cell Biology Center, Institute of Innovative Research, Tokyo Institute of Technology, Kanagawa, Japan, **3** Department of Molecular Cell Biology, Graduate School of Comprehensive Human Sciences and Institute of Basic Medical Sciences, University of Tsukuba, Ibaraki, Japan, **4** Live Cell Super-Resolution Imaging Research Team, RIKEN Center for Advanced Photonics, Saitama, Japan, **5** Laboratory of Glycobiology and Glycotechnology, Department of Applied Biochemistry, School of Engineering, Tokai University, Kanagawa, Japan, **6** Key Laboratory of Carbohydrate Chemistry and Biotechnology, Ministry of Education, School of Biotechnology, Jiangnan University, Wuxi, China, **7** Laboratory for Cell Dynamics Observation, Center for Biosystems Dynamics Research, RIKEN, Osaka, Japan, **8** Department of Physics and Universal Biology Institute, Graduate School of Science, The University of Tokyo, Tokyo, Japan, **9** Department of Physics, Universal Biology Institute, and the International Research Center for Neurointelligence (WPI-IRCN), The University of Tokyo, Tokyo, Japan, **10** Department of Biochemistry and Cell Biology, Stony Brook University, Stony Brook, New York, United States of America, **11** Collaborative Research Institute for Innovative Microbiology, The University of Tokyo, Tokyo, Japan

* atachi@g.ecc.u-tokyo.ac.jp

## Abstract

Vps13 family proteins are proposed to function in bulk lipid transfer between membranes, but little is known about their regulation. During sporulation of *Saccharomyces cerevisiae*, Vps13 localizes to the prospore membrane (PSM) via the Spo71–Spo73 adaptor complex. We previously reported that loss of any of these proteins causes PSM extension and subsequent sporulation defects, yet their precise function remains unclear. Here, we performed a genetic screen and identified genes coding for a fragment of phosphatidylinositol (PI) 4-kinase catalytic subunit and PI 4-kinase noncatalytic subunit as multicopy suppressors of *spo73Δ*. Further genetic and cytological analyses revealed that lowering PI4P levels in the PSM rescues the *spo73Δ* defects. Furthermore, overexpression of *VPS13* and lowering PI4P levels synergistically rescued the defect of a *spo71Δ spo73Δ* double mutant, suggesting that PI4P might regulate Vps13 function. In addition, we show that an N-terminal fragment of Vps13 has affinity for the endoplasmic reticulum (ER), and ER-plasma membrane (PM) tethers localize along the PSM in a manner dependent on Vps13 and the adaptor complex. These observations suggest that Vps13 and the adaptor complex recruit ER-PM tethers to ER-PSM contact sites. Our analysis revealed that involvement of a phosphoinositide, PI4P, in regulation of Vps13, and also suggest that distinct contact site proteins function cooperatively to promote *de novo* membrane formation.

**Data Availability Statement:** All relevant data are within the manuscript and its Supporting Information files.

**Funding:** This work was supported by JSPS KAKENHI grant 17K07712 and 20K05782 to H. Tachikawa, grants from the Noda Institute for Scientific Research (https://www.nisr.or.jp/english/) and Asahi Group Foundation (http://www.asahigroup-foundation.com/academic/) to H.Tachikawa, NIH grant GM072540 to A. Neiman, and a Grant-in-Aid for JSPS Fellows 17J06832 to T.S.Nakamura. The funders had no role in study design, data collection and analysis, decision to publish, or preparation of the manuscript.

**Competing interests:** The authors have declared that no competing interests exist.

## Author summary

Vps13 family proteins are conserved lipid transfer proteins that function at organelle contact sites and have been implicated in a number of different neurological diseases. In the yeast *Saccharomyces cerevisiae*, Vps13 is encoded by a single gene and is localized to various contact sites by interaction with different adaptor proteins and/or lipids, however its regulation is yet to be clarified. We have previously shown that during the developmental process of sporulation, Vps13 is recruited to *de novo* membrane structures called prospore membranes (PSMs) by a specific adaptor complex, and Vps13 and its adaptors are required for PSM extension. Here we reveal that loss of an adaptor can be overcome by lowering phosphatidylinositol-4-phosphate (PI4P) levels, either by inhibiting PI 4-kinase on the PSM or recruiting PI 4-phospatase to the PSM and that PI4P levels in the PSM affect Vps13 function. Further, we show that Vps13 forms endoplasmic reticulum (ER)-PSM contact sites, that ER-plasma membrane tethering proteins are recruited to ER-PSM contacts, and these proteins may function in conjunction with Vps13. Thus, our work shines light on both the mechanisms of intracellular remodeling and the function of this important class of lipid transfer proteins.

## Introduction

Phosphoinositides (PIPs) are minor components of cellular phospholipids that play key roles in cellular functions such as signal transduction, cytoskeletal organization, membrane trafficking, and the establishment and maintenance of organelle identity [1]. Among PIPs, phosphatidylinositol-4-phosphate (PI4P) participates in the secretory pathway, wherein it creates membranous structures at the Golgi apparatus and recruits various effector proteins to facilitate post-Golgi membrane trafficking [2–4]. It is also an important constituent of the plasma membrane (PM). In yeast, PI4P is synthesized at the Golgi and the PM by distinct PI 4-kinases (PI4K), Pik1 and Stt4, respectively. Pik1 (a homolog of mammalian PI4KIIIβ) localizes at the Golgi and promotes vesicle trafficking, whereas Stt4 (a homolog of mammalian PI4KIIIα) forms a PI4K complex with Efr3 and Ypp1 that is essential for cellular growth and lipid homeostasis at the PM [5–9].

In the last decade, an intriguing concept has been established that, at membrane contact sites (MCSs), where two organelle membranes are closely apposed, various lipids are transported by lipid transfer proteins (LTPs) [10]. Yeast Osh proteins, mammalian oxysterol-binding proteins (OSBP), and OSBP-related proteins (ORP) are major LTPs, and they countertransport PI4P and phosphatidylserine or sterol through a shuttle mechanism [11–13]. In addition, SMP domain-containing proteins such as yeast tricalbin [14] or mammalian extended-synaptotagmin (E-Syt) [15] transport lipid molecules between organelles in a similar manner.

More recent structural studies of two distantly related proteins, Atg2 and Vps13, suggest an alternate mechanism for lipid transport at contact sites [16,17]. Vps13 was originally characterized for its role in membrane traffic [18–20] and yeast sporulation [21–23]. In recent years, however, Vps13 has been revealed to associate with membrane-specific adaptors, thereby localizing to various MCSs, including contacts between the nucleus and vacuoles (NVJ), mitochondria and vacuoles (vCLAMP), and endosomes and the mitochondria [24–27]. Structural studies of a fungal Vps13 N-terminal region have revealed that Vps13 forms a long hydrophobic channel that can accommodate multiple lipid molecules, suggesting that Vps13 can mediate bulk transport of lipids between membranes [17,28]. Atg2, which shares some sequence

homology with Vps13 and is essential for the formation of isolation membranes in autophagy [29], forms a similar hydrophobic channel [30], suggesting that this mechanism of bulk lipid transport is important for supplying phospholipids for organelle morphogenesis. Despite such advances, how Vps13 functions during sporulation of budding yeast has not been settled.

Sporulation is the process of gametogenesis in which four haploid nuclei produced by meiosis are engulfed by a *de novo* membrane structure, called the prospore membrane (PSM), and this is followed by spore maturation [31]. Since the source of the PSM is post-Golgi vesicles, in mutants defective in post-Golgi membrane trafficking clustered post-Golgi vesicles are observed instead of the nascent PSM [32,33]. Interestingly, visual screening of PSM morphology has identified mutants defective in the extension of the PSM, in which PSMs are formed but do not extend properly. Mutations in *VPS13*, *GIP1*, *SPO71*, and *SPO73* all display this phenotype [21,23,34,35]. Gip1 is a subunit of protein phosphatase type 1 (PP1) and functions with the catalytic subunit Glc7, while Vps13 functions in concert with Spo71 and Spo73 [23,34,36,37].

Spo71 is a protein with three pleckstrin homology (PH) domains that bind lipids or proteins [38]. Spo71 directly interacts with Vps13, recruits Vps13 to the PSM and *spo71Δ/spo71Δ* (hereafter *spo71Δ*) mutants display similar sporulation defects to *vps13Δ/vps13Δ* (hereafter *vps13Δ*) mutants [23,26]. Spo73 is a small dysferlin domain-only protein. Dysferlin domains are conserved in a subset of ferlin proteins, such as dysferlin, which is involved in PM repair in muscle cells [39], implying that the dysferlin domain might contribute to membrane trafficking or lipid homeostasis. Spo71 and Spo73 are expressed specifically during sporulation, colocalize on the PSM, and interact with each other [34,37], thus Spo71-Spo73 can be considered as an adaptor complex for Vps13 on the PSM. However, the precise mechanism and regulation of PSM extension by Vps13 and the Spo71-Spo73 adaptor complex is not understood.

In this study, we identify a truncated allele of *STT4* and *EFR3* as multicopy suppressors of *spo73Δ/spo73Δ* (hereafter *spo73Δ*) and show that lowering PI4P levels in the PSM suppresses the defects of *spo73Δ*. Lowering PI4P levels can also partially suppress *spo71Δ*, but not *vps13Δ*, suggesting that PI4P may negatively regulate the Vps13 channel. We also show that tethers for endoplasmic reticulum (ER)-PM contact sites localize to the PSM, dependent on Vps13 and the adaptor complex. These results suggest that distinct tethering complexes cooperate to form ER-PSM contact sites that permit Vps13 to contribute lipids for PSM extension.

## Results

### Overexpression of *EFR3* or a fragment of *STT4* gene partially suppress the sporulation defect of *spo73Δ*

To investigate the role of Spo73, a subunit of the Vps13 adaptor complex, we screened for multicopy suppressors of *spo73Δ*. The *spo73Δ* cells were transformed with a yeast genomic library, sporulated, and subjected to ethanol treatment. Only mature spores are resistant to ethanol, while vegetative cells and immature or aberrant spores are sensitive. Clones that could form ethanol-resistant spores were isolated, and genomic regions responsible for suppression were identified. In this screening, *EFR3* and fragments of *STT4* (1–814) (hereinafter referred to as *STT4frag*) were obtained. To assess the sporulation efficiency, we observed mature spores visible under light microscope. While *spo73Δ* cells showed no sporulation, *spo73Δ* cells overexpressing *STT4frag* or *EFR3* showed 17.8% and 9.8% sporulation, respectively (Fig 1A). Stt4 and Efr3 form the PI 4-kinase (PI4K) complex with Ypp1 on the PM (Fig 1B) [7,8]; Efr3 tethers the PI4K complex to the PM via N-terminal basic residues, and Ypp1 links Stt4 and Efr3. Thus, our results indicate that *SPO73* has a genetic interaction with genes for the PM PI4K complex.

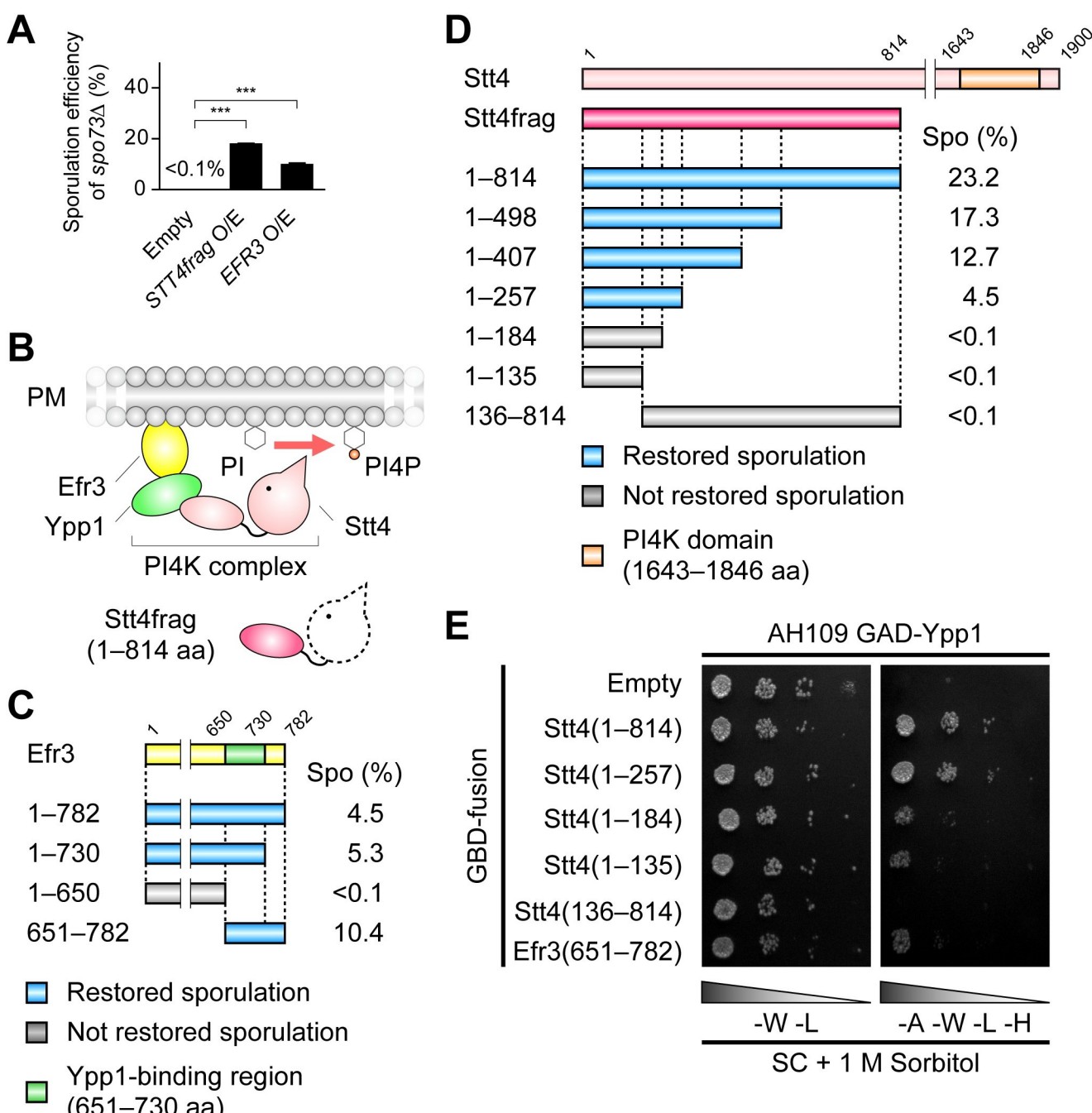

**Fig 1. Overexpression of *EFR3* or truncated *STT4* partially suppress the sporulation defect of *spo73Δ*.** (A) Percentage of asci in *spo73Δ* (TC545) overexpressing *STT4frag*, *EFR3*, or harboring empty vector (control). More than 200 cells were observed in three independent colonies of each strain harboring indicated plasmids (for a total of > 600 cells). The bar graph shows the mean of the percentage of asci (N = 3). ***, p < 0.001 (Dunnett's test). (B) Diagram of PI4K complex on the PM. Stt4frag is an N-terminal fragment of Stt4 (1–814 aa). (C) Assessment of sporulation of *spo73Δ* (TC545) by overexpression of constructs encoding a Efr3 deletion series. Spo (%) indicates the percentage of asci containing refractile spore(s). Light blue and gray boxes indicate deletion mutants that restored sporulation, or not, respectively. The green box indicates the Ypp1-binding region (651–730 aa). (D) Assessment of sporulation of *spo73Δ* (TC545) by overexpression of constructs encoding a Stt4 deletion series. Diagram is shown as in (C). Orange box indicates PI4K domain (1643–1846 aa). (E) Assessment of binding of Ypp1 and Stt4 fragments by yeast two-hybrid assay. AH109 cells producing GAD-Ypp1 and indicated GBD-fusion proteins were cultured, and 10-fold serial dilutions were spotted onto SC plates depleted of tryptophan and leucine (-W -L), or depleted of adenine, tryptophan, leucine and histidine (-A -W -L -H) containing 1 M sorbitol for osmotic adjustment. Each plate was incubated for 2 days at 30°C. Binding of Ypp1 and a Efr3 fragment was assessed as a positive control for functionality of GAD-Ypp1.

## The PI4K complex localizes on the PSM, and suppression occurs through a dominant-negative effect on this complex

Stt4 is a catalytic subunit of the PI4K complex; however, Stt4frag lacks the kinase domain. To identify the regions of *STT4frag* and *EFR3* important for suppression, a deletion series of these genes was constructed and overexpressed in *spo73Δ* cells. For *EFR3*, deletion causing lack of the C-terminal Ypp1-binding region (residues 651–730) could not suppress *spo73Δ* (Fig 1C), suggesting that binding to Ypp1 is important for suppression. In the *STT4frag* deletion series, fragment encoding Stt4(1–257) was sufficient for suppression (Fig 1D). Because Stt4 has been shown to bind to Ypp1 via the N-terminal and middle regions [7], we tested the capacity of the Stt4frag and its parts to bind to Ypp1 by a yeast two-hybrid assay. The fragments, that suppressed *spo73Δ*, including Stt4(1–257), bound more strongly to Ypp1 than fragments that could not suppress *spo73Δ* (Stt4(1–184) and Stt4(1–135)) (Fig 1E). These results suggest that overexpression of *STT4frag* or *EFR3* affects the formation of the PI4K complex. The degree of suppression varied somewhat between different *STT4* and *ERF3* constructs, likely due to differences in expression level, though this has not been directly examined.

Considering the lack of a kinase domain in Stt4frag, we presumed that overexpression of *STT4frag* might negatively affect the function of the PI4K complex. To test this hypothesis, we constructed a plasmid encoding a kinase-dead (KD) Stt4 (carrying a D1754A substitution) based on the mutant Pik1 [40], and confirmed its inactivity (S1A Fig). When either wild-type *STT4* (WT) or *STT4-KD* were overexpressed in *spo73Δ* cells, only *STT4-KD* suppressed *spo73Δ* (4.3% sporulation) (S1B Fig), suggesting that overexpression of *STT4frag* or *STT4-KD*, encoding inactive kinase, inhibited the formation of the active PI4K complex on the PSM in a dominant-negative manner, leading to the suppression of *spo73Δ*.

We next examined the localization of the PI4K complex during sporulation in wild-type and *spo73Δ* cells. In either cells, GFP-Stt4 appeared as discrete puncta along the PSM, marked by mRFP-Spo20$^{51-91}$, mRFP fused to the sequence of Spo20 which binds phosphatidic acid on the PSM (Fig 2A and 2B). Ypp1 and Efr3 also localized on the PSM, while Sfk1, a transmembrane protein at the PM which is reported to bind to Stt4 [41], did not localize to the PSM (Figs 2C, 2D, S1C, and S1D). How might *EFR3* overexpression inhibit the function of Stt4 at the PSM? In sporulating cells Efr3-GFP expressed from a low copy plasmid was found faintly in patches along the PSM, similar to GFP-Stt4 (Fig 2C and 2D). Upon overproduction, Efr3-GFP could be seen uniformly along the PSM but was now predominantly on the ascal PM. We then tested the effect of *EFR3* overexpression on the localization of Stt4. Stt4 was relocated from the PSM to the PM (Fig 2E), suggesting that overproduced Efr3 sequesters Stt4 from the PSM. Collectively, these results suggest that overexpression of *STT4frag* and *EFR3* cause suppression of *spo73Δ* through an inhibitory effect on the PI4K complex at the PSM.

## Sporulation-specific depletion of Stt4 suppresses the membrane extension defect of *spo73Δ*

The results above suggest that reduced activity of the Stt4 complex at the PSM suppresses *spo73Δ*. To confirm this, we tried to perturb Stt4 function. Because all the components of the PI4K complex, Stt4, Ypp1, and Efr3, are essential for growth, we repressed the expression of *STT4* specifically during sporulation using the promoter of *CLB2* [42,43] and fusing sequence coding an auxin-inducible degron (AID) tag at the N-terminus of Stt4 [44,45]. In addition, to effectively synchronize meiosis and sporulation, we introduced an *NDT80* block/release (*NDT80*-B/R) system, in which expression of *NDT80*, a master regulator of the middle genes of sporulation, is controlled by an estrogen-inducible promoter (Fig 3A) [46,47]. We induced sporulation of *spo73Δ NDT80*-B/R P$_{CLB2}$-*degron-Stt4* cells and found that these cells showed

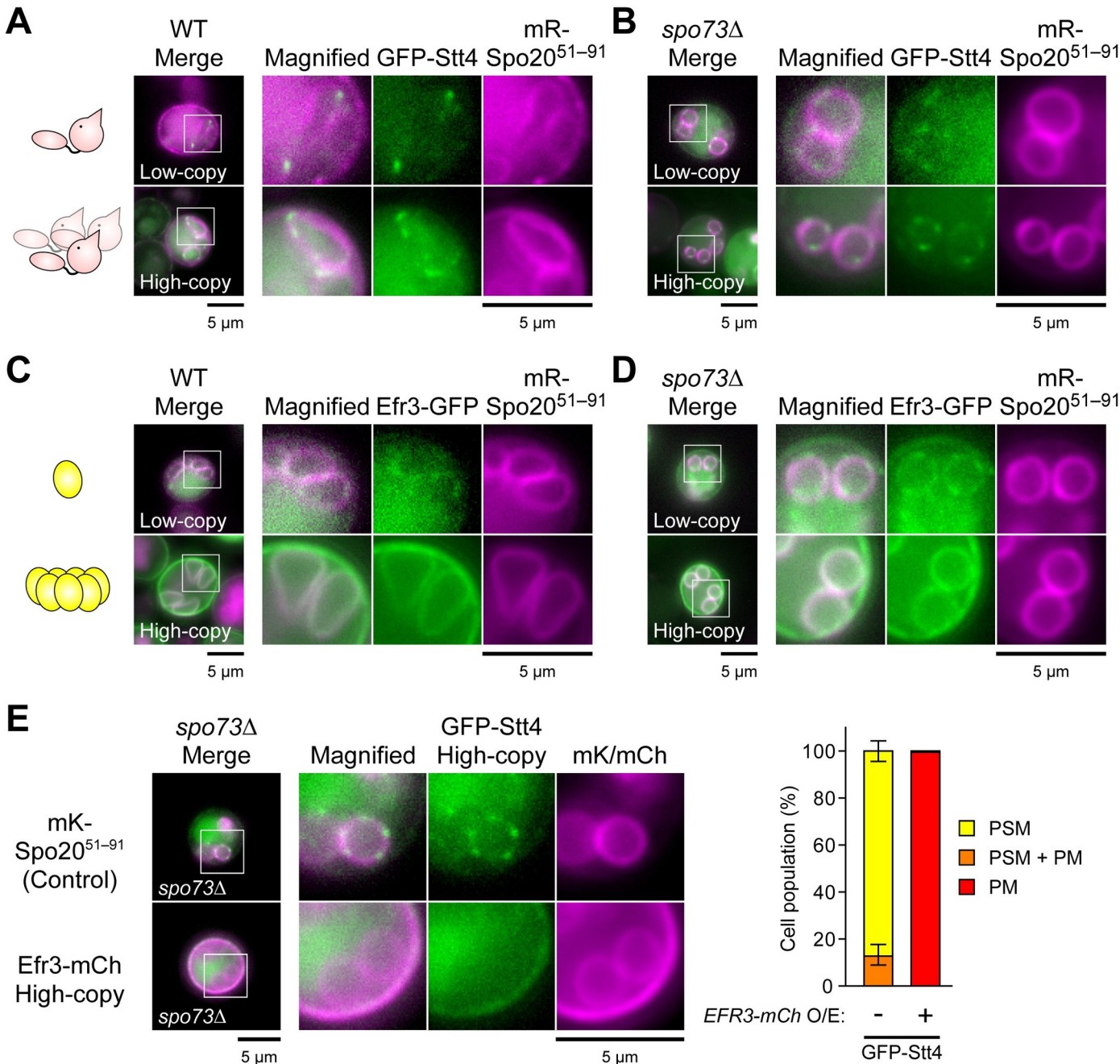

**Fig 2. The PI4K complex localizes on the PSM, and suppression occurs through a dominant-negative effect.** (A–D) Localization of Stt4 (A and B) and Efr3 (C and D) in wild-type (AN120, A and C) or *spo73Δ* (TC545, B and D) cells during PSM formation. In each panel, *GFP-STT4* and *EFR3-GFP* were expressed from low-copy (upper) or high-copy vector (lower). mR, mRFP. mR-Spo20$^{51–91}$, a PSM marker. (E) Assessment of localization of overproduced GFP-Stt4 in *spo73Δ* (TC545) cells producing mKate2-Spo20$^{51–91}$ (control), or overproducing Efr3-mCherry during PSM formation. Left: Representative images. Right: Percentage of cells in which GFP-Stt4 localized only on the PSM (PSM), both on the PSM and PM (PSM + PM), or only on the PM (PM). Twenty cells were observed in three independent colonies of each strain harboring indicated plasmids (for a total of > 60 cells). The bar graph shows mean ± SEM of the percentage of cells (N = 3). mK, mKate2. mCh, mCherry. Scale bar, 5 μm.

some sporulation even in auxin-absent conditions (Fig 3B). The amount of degron-Stt4 was reduced in nutrient-starved conditions, confirming that the expression of *STT4* was blocked by the *CLB2* promoter (Fig 3C). Furthermore, the expression of *GFP-STT4* from the

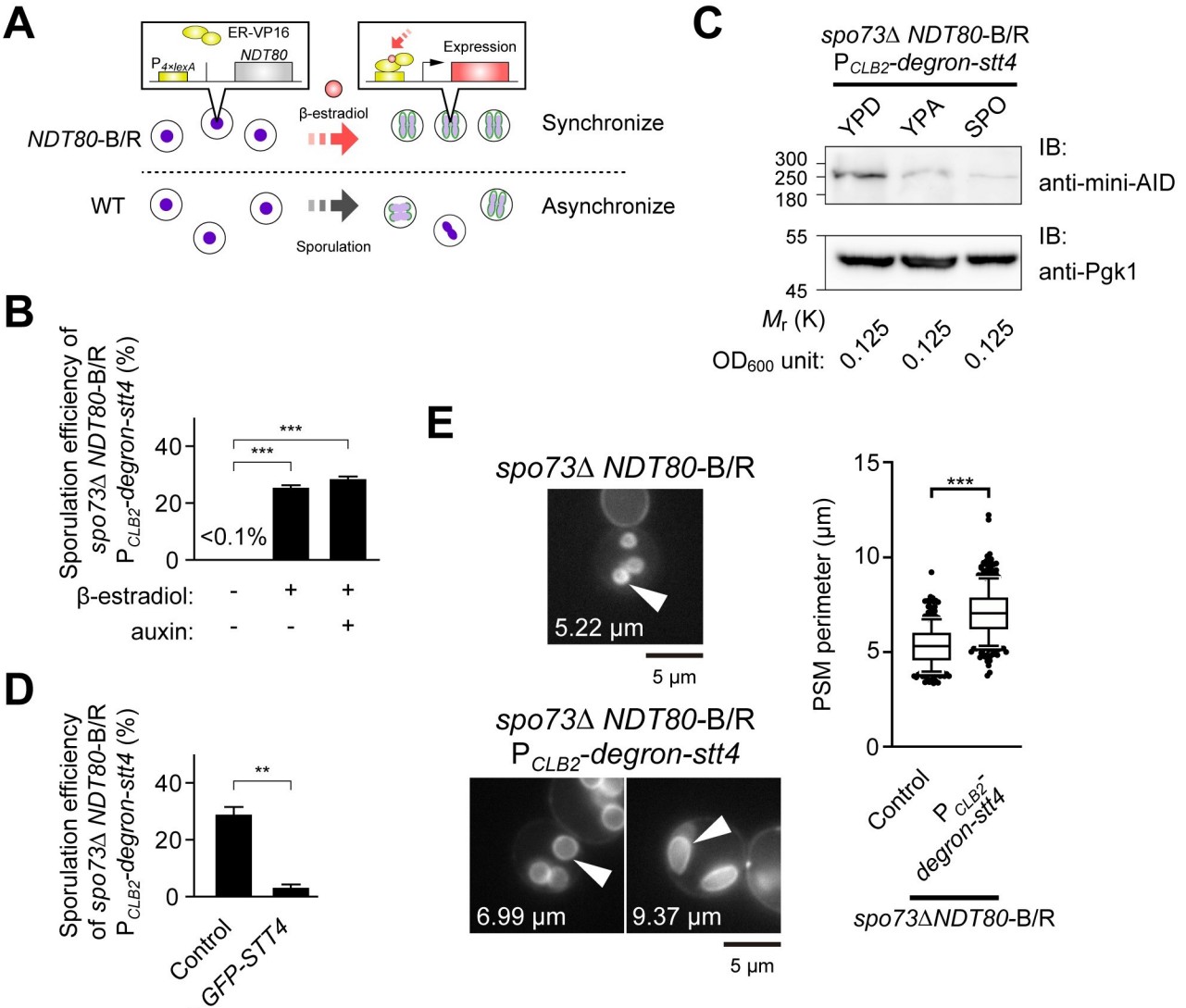

**Fig 3. Sporulation-specific depletion of Stt4 suppresses the membrane extension defect of *spo73Δ*.** (A) Diagram of *NDT80* block/release (*NDT80*-B/R) system. ER-VP16 is a chimeric protein of the bacterial DNA-binding protein LexA, the hormone-binding domain of the human estrogen receptor (ER) and fragment of the herpes simplex virus type 1 trans-activator VP16. When β-estradiol binds ER, ER-VP16 translocates into nucleus, binds $P_{4×lexA}$ sequence, and induces expression of downstream coding sequence. (B) Assessment of sporulation in *spo73Δ NDT80*-B/R $P_{CLB2}$-*degron-stt4* (TNY502) with β-estradiol and auxin. More than 200 cells were observed in three independent colonies of each condition (for a total of > 600 cells). The bar graph shows the mean of the percentage of asci (N = 3). ***, p < 0.001 (Tukey-Kramer test). (C) Assessment of protein level of degron-Stt4 in *spo73Δ NDT80*-B/R $P_{CLB2}$-*degron-stt4* (TNY502) cells cultured in YPD, YPA, or SPO medium. The degron-Stt4 and Pgk1, as an internal control, were detected by anti-mini-AID and anti-Pgk1, respectively. Amounts of cells were adjusted to 0.125 $OD_{600}$ units. (D) Assessment of sporulation in *spo73Δ NDT80*-B/R $P_{CLB2}$-*degron-stt4* expressing *GFP-STT4* from $P_{ADH1}$ (TNY517). TNY502 was assessed as a control strain. More than 200 cells were observed in three independent colonies of each condition (for a total of > 600 cells). The bar graph shows means of the percentage of asci (N = 3). **, p < 0.01 (Student's t test). (E) Left: Representative cells are shown. The perimeter of the PSM indicated by white arrow heads are shown. Right: Assessment of the perimeter of the PSM in *spo73Δ NDT80*-B/R producing mKate2-Spo20$^{51–91}$ (TNY198) or *spo73Δ NDT80*-B/R $P_{CLB2}$-*degron-stt4* producing mKate2-Spo20$^{51–91}$ (TNY518) cells. More than 50 PSMs were measured in three independent colonies of each strain (for a total of 250 or 303 PSMs, respectively). The whiskers indicate the 10th and 90th percentiles and dots indicate outliers. ***, p < 0.001 (Student's t test). mK, mKate2. Scale bar, 5 μm.

constitutive *ADH1* promoter again repressed sporulation of *spo73Δ NDT80*-B/R $P_{CLB2}$-*degron-Stt4* cells, indicating that the downregulation of Stt4 suppressed the defect of *spo73Δ* (Fig 3D). Measurements of the PSM perimeter showed that the PSM of *spo73Δ NDT80*-B/R $P_{CLB2}$-

*degron-Stt4* cells was significantly larger than that of *spo73Δ NDT80*-B/R cells (Fig 3E). These results indicate that PSM extension is also restored in the process of the suppression of *spo73Δ*.

## Selective depletion of PI4P in the PSM suppresses the defects of *spo73Δ*

Given that the Stt4 complex is likely responsible for the PI4P pool in the PSM, we hypothesized that reducing PI4P in the PSM by other means would cause suppression of the *spo73Δ* phenotype. In mammals, selective depletion of PI4P on the PM can be accomplished by targeting heterologously expressed PI4P 4-phosphatase domain of yeast Sac1 (hereinafter referred to as Sac1$^{2-517}$) [48]. Accordingly, we constructed genes encoding Sac1$^{2-517}$ fused with two PSM marker proteins, mRFP-Spo20$^{51-91}$, which recognizes phosphatidic acid on the PSM, or Dtr1, a PSM-resident transmembrane protein (Fig 4A) [49,50]. Both phosphatase-active (WT) and phosphatase-dead (PD) chimera proteins were localized on the PSM, confirming that these targeting signals work (S2 Fig). While full length Sac1 and the phosphatase-dead chimeras in *spo73Δ* cells did not show suppression, overproduction of the phosphatase-active chimeras suppressed *spo73Δ* phenotype. (Fig 4B, mRFP-Spo20$^{51-91}$: 15.4% sporulation, mRFP-Dtr1: 7.2%). In addition, overexpression of a 3×mKate2-Spo20$^{51-91}$-Sac1$^{2-517}$ chimera using an estrogen-inducible promoter in *spo73Δ NDT80*-B/R cells showed that the PSM was extended compared to control cells (Fig 4C). These results strongly suggest that reduction of the PI4P level in the PSM is the basis for suppression of the *spo73Δ* sporulation defect.

## Changes in PI4P levels in the PSM can be detected using an improved PI4P biomarker

Although overproduction of Sac1$^{2-517}$ chimeras suppressed the defects of *spo73Δ*, we could not detect a decrease in PI4P levels in the PSM using PH$^{Osh2}$, a classical PI4P biomarker (Fig 5A) [51]. Both single and tandem PH$^{Osh2}$ localized on the PSM, even in cells overexpressing Sac1$^{2-517}$ chimera proteins (S3A Fig). No Golgi localization of the reporters was seen in sporulating cells. Given that the tandem PH$^{Osh2}$ localizes both on the Golgi and on the PM in the bud during vegetative growth [51], exclusive localization of PH$^{Osh2}$ to the PSM implies that PH$^{Osh2}$ has high affinity to the PSM during PSM formation. To test this, we also localized a reporter using P4M-SidM, a PI4P-binding protein derived from the bacterial pathogen *Legionella pneumophila* (hereinafter referred to as P4M) [52]. In contrast to PH$^{Osh2}$, P4M localized preferentially to Sec7-positive *trans*-Golgi structures during vegetative growth (S3B Fig) and in either wild-type or *spo73Δ* cells during PSM formation (S3C and S3D Fig). This suggests that in contrast to mammalian cells, in which P4M behaves as unbiased PI4P marker [52], P4M prefers PI4P on the Golgi during PSM formation in budding yeast.

Hence, we constructed a fusion of both PH$^{Osh2}$ and P4M with GFP (referred to as Osh2-P4M) and found that Osh2-P4M localized at both the PM and Golgi during vegetative growth in wild-type cells (S4A and S4B Fig). To examine dependency of Osh2-P4M localization on PI4P, we used temperature-sensitive (ts) mutants of *STT4* and *PIK1*, which encodes Golgi-localized PI4K. When *stt4-ts* and *pik1-ts* are incubated at nonpermissive temperature, the PI4P pool of the PM and the Golgi are depleted, respectively. Under this condition, Osh2-P4M did not localize to the PM in *stt4-ts*, and to the Golgi in *pik1-ts* (S4C Fig). During sporulation, Osh2-P4M localized on both the PSM and the Golgi during PSM formation (Figs 5B and S4D). We suppose that distribution of Osh2-P4M between the PSM or the Golgi in a given cell is a response to the relative PI4P levels in those organelles, which allowed us to test the decrease of PI4P in the PSM during suppression of *spo73Δ*. In *spo73Δ NDT80*-B/R cells, the distribution of Osh2-P4M localization was 43.4% cells with PSM only, 54.5% displayed both PSM and Golgi localization, and 2.1% only the Golgi (Fig 5C and 5D). Upon the

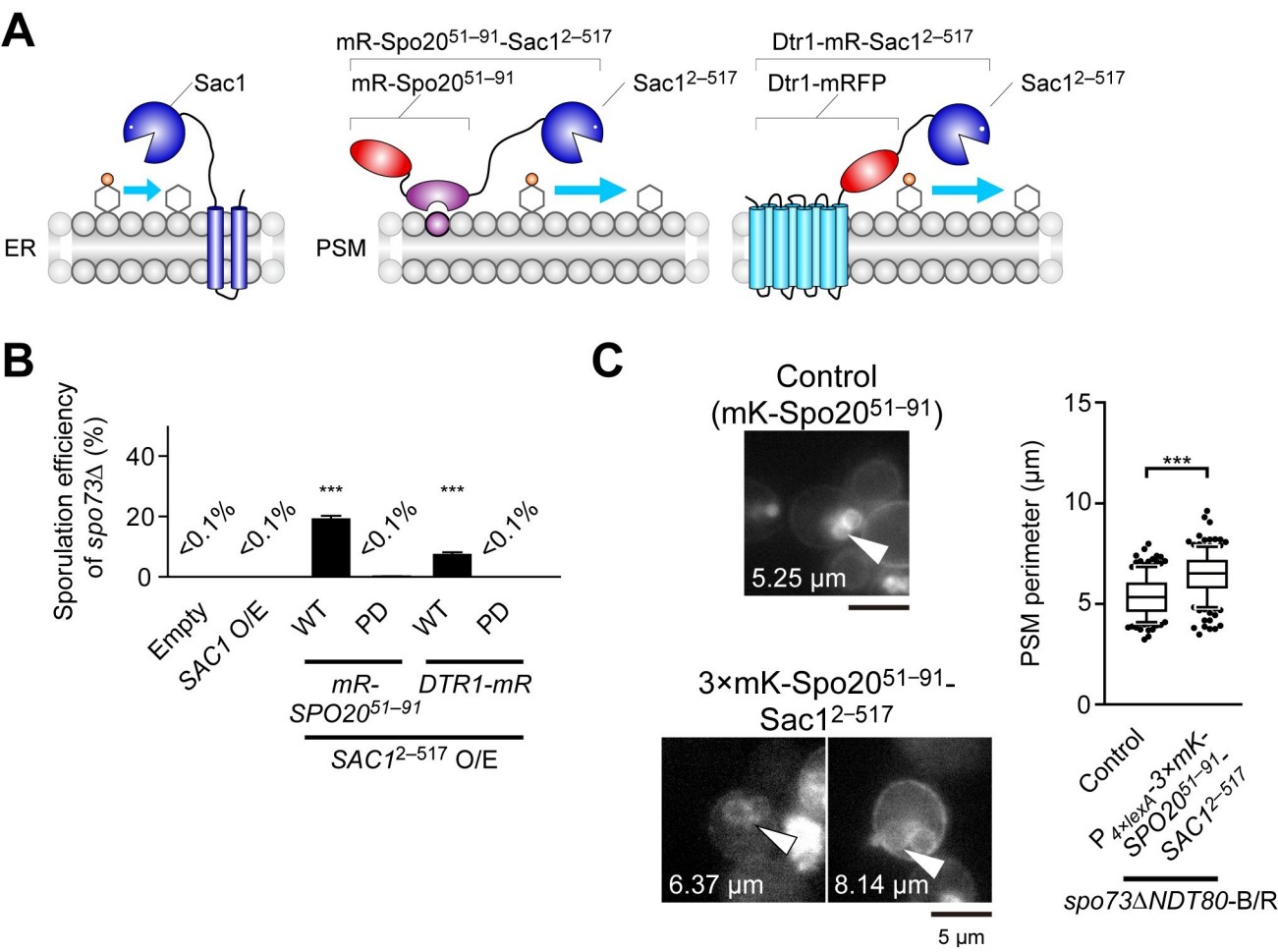

**Fig 4. Selective depletion of PI4P in the PSM suppresses the defects of *spo73Δ*.** (A) Diagram of full length Sac1 and Sac1$^{2-517}$ chimera proteins. Sac1$^{2-517}$: catalytic domain of yeast Sac1 (2–517 aa). See text for details. mR, mRFP. (B) Assessment of sporulation in *spo73Δ* (TC545) overexpressing constructs encoding full length Sac1, phosphatase active (WT) or phosphatase-dead (PD) Sac1$^{2-517}$ chimera proteins. More than 200 cells were observed in three independent colonies of each strain harboring indicated plasmids (for a total of > 600 cells). The bar graph shows the mean of the percentage of asci (N = 3). ***, p < 0.001 (Tukey-Kramer test). (C) Left: Representative cells are shown. The perimeter of the PSM indicated by white arrow heads are shown. Right: Assessment of perimeter of the PSM in *spo73Δ NDT80*-B/R producing mKate2-Spo20$^{51-91}$ (TNY198), as a control, or 3×mKate2-Spo20$^{51-91}$-Sac1$^{2-517}$ from β-estradiol inducible promoter (TNY578). More than 50 PSMs were measured in three independent colonies of each strain (for a total of 151 or 152 PSMs, respectively). The whiskers indicate the 10th and 90th percentiles and dots indicate outliers. ***, p < 0.001 (Student's t test). mK, mKate2. Scale bar, 5 μm.

expression of gene fusions encoding active Sac1$^{2-517}$ chimeras, the population of cells displaying only PSM localization decreased to 10.3% in *spo73Δ NDT80*-B/R cells, indicating a decrease in PI4P levels in the PSM, while expression of those encoding inactive Sac1$^{2-517}$ (Sac1$^{2-517}$-PD) did not alter the distribution of Osh2-P4M localization pattern. Collectively, these results indicate that a decrease in PI4P levels in the PSM suppressed the defect of spore formation and PSM extension in *spo73Δ*.

## The PSM is distinctively abundant in PI4P

Since PI4P is the precursor of PI(4,5)P$_2$, which has been implicated in PSM formation [53], the effects of changes in PI4P level that we see could be mediated indirectly through changes in PI(4,5)P$_2$ levels. Considering earlier reports showing that the PSM can be labelled with marker

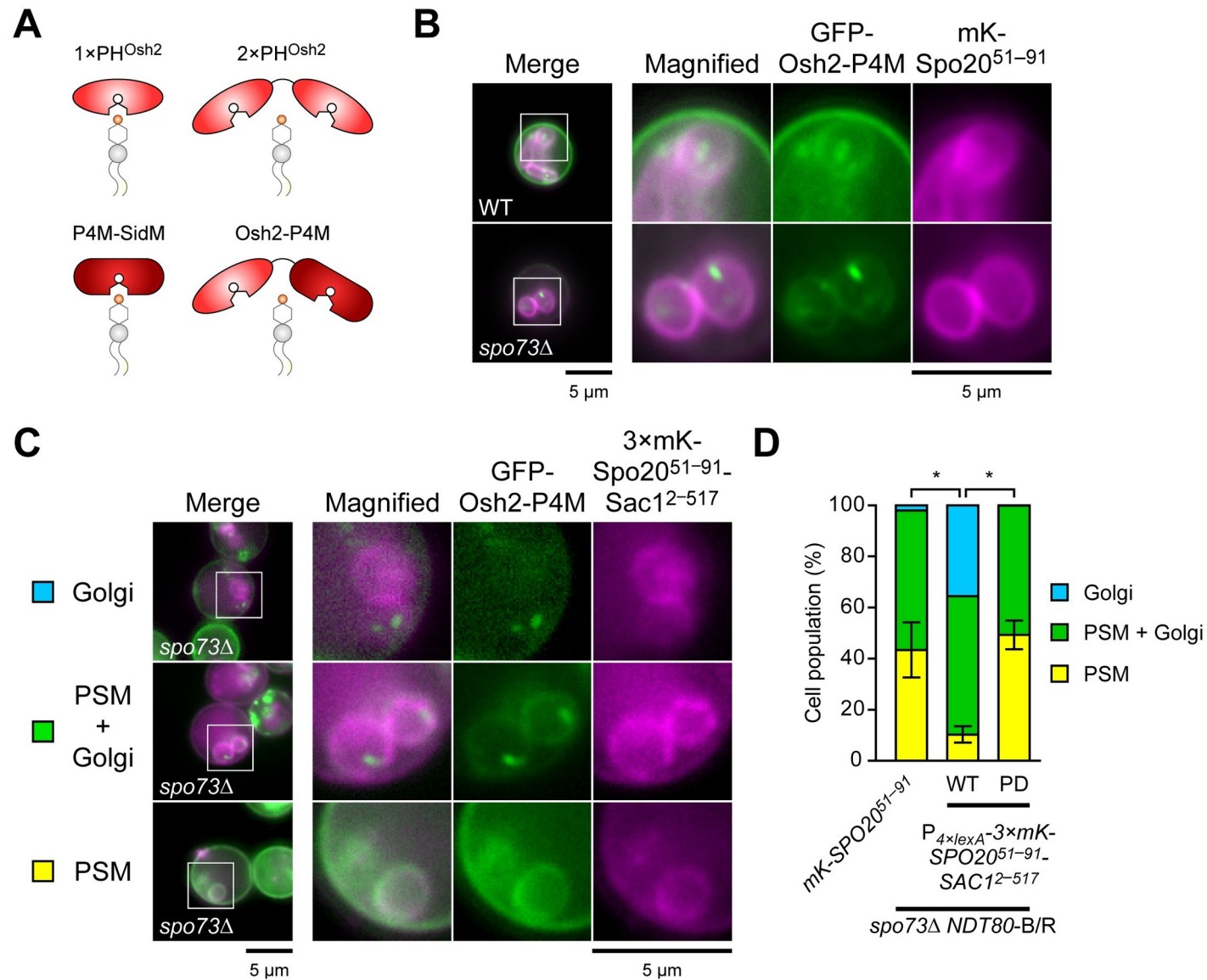

**Fig 5. Changes in PI4P levels in the PSM can be detected using an improved PI4P biomarker.** (A) Diagram of PI4P markers used in this study. See text for details. (B) Localization of GFP-Osh2-P4M in wild-type (TNY375) and *spo73*Δ (TNY376) cells during PSM formation. (C) Representative localization of GFP-Osh2-P4M in *spo73*Δ *NDT80*-B/R cells producing 3×mKate2-Spo20$^{51–91}$-Sac1$^{2–517}$ during PSM formation. (D) Assessment of localization of GFP-Osh2-P4M in *spo73*Δ *NDT80*-B/R cells expressing indicated constructs. More than 100 cells were observed in three independent colonies of each condition. The bar graph shows mean ± SEM of the percent of the PSM population of each condition (N = 3). *spo73*Δ *NDT80*-B/R cells expressing GFP-Osh2-P4M and mKate2-Spo20$^{51–91}$ (TNY403), 3×mKate2-Spo20$^{51–91}$-Sac1$^{2–517}$ (TNY421), or 3×mKate2-Spo20$^{51–91}$-Sac1$^{2–517}$-PD (TNY429) were observed. *, p < 0.05 (Tukey-Kramer test). mK, mKate2. mKate2-Spo20$^{51–91}$, a PSM marker. Scale bar, 5 μm.

proteins for both PI4P and PI(4,5)P$_2$ [22,54], we examined the distribution of different phosphoinositides during sporulation in wild-type and *spo73*Δ cells. As shown above, PI4P was distributed on both the growing and mature PSM in wild-type cells and even in *spo73*Δ cells (Fig 6A). We additionally observed the distributions of PI3P and PI(4,5)P$_2$ using canonical lipid markers during PSM formation. FYVE$^{EEA1}$, a PI3P marker, localized on endosome-like puncta and vacuolar membranes (Fig 6B). As for PI(4,5)P$_2$, we observed that 2×PH$^{PLCδ}$, a PI(4,5)P$_2$ marker, remained on the PM during the extension and after the maturation of the PSM (Fig 6C, top and middle). In post-meiotic cells, however, 2×PH$^{PLCδ}$ appeared on the mature PSM,

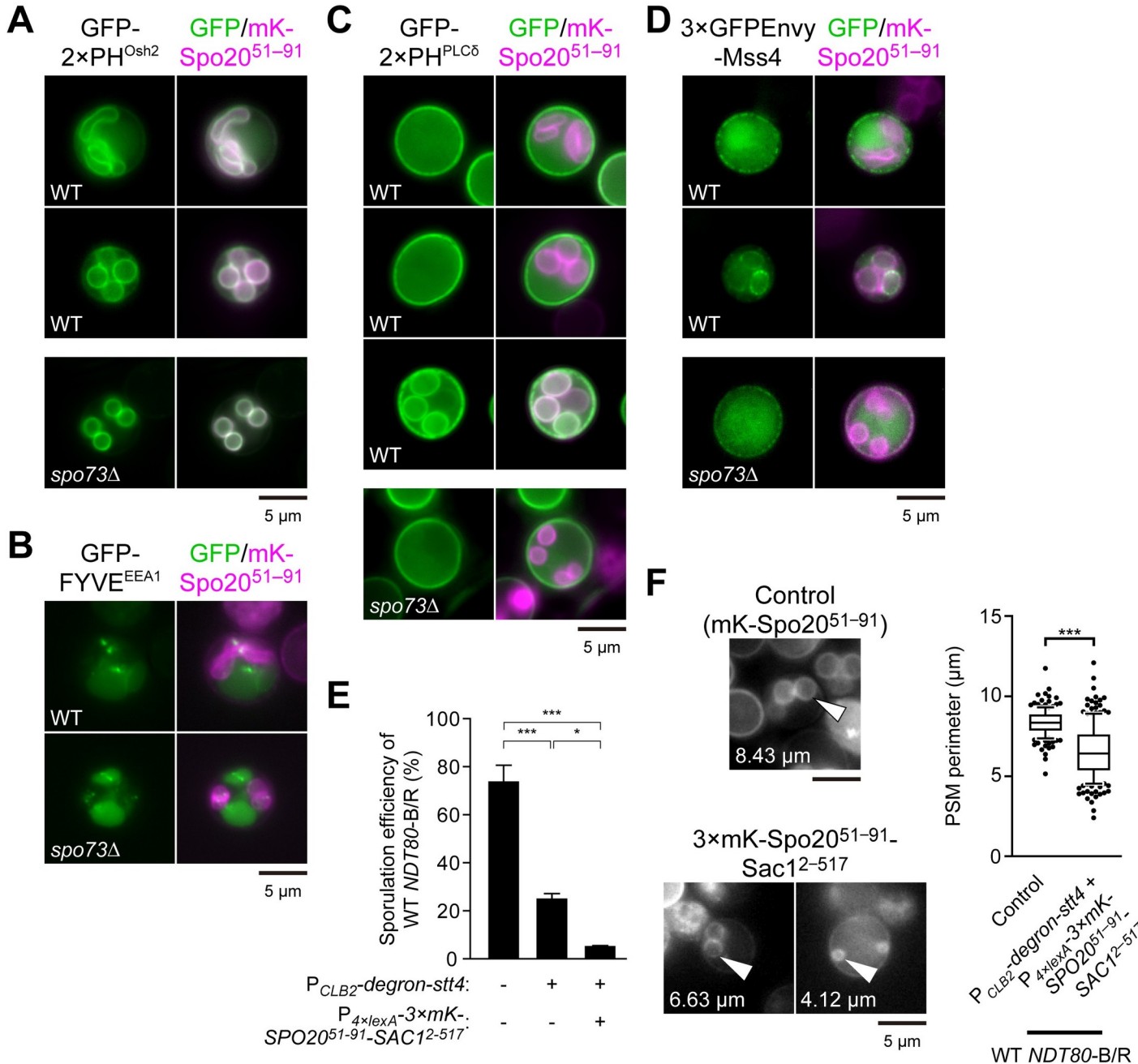

**Fig 6. The PSM is distinctly abundant in PI4P.** (A) Localization of GFP-2×PH$^{Osh2}$ in wild-type (AN120) cells during PSM extension (top) or after PSM closure (bottom), and *spo73*Δ (TC545) cells during PSM formation. (B) Localization of GFP-FYVE$^{EEA1}$ in wild-type (AN120) and *spo73*Δ (TC545) cells during PSM formation. (C) Localization of GFP-2×PH$^{PLCδ}$ in wild-type (AN120) cells during PSM extension (top) or after PSM closure (middle and bottom), and *spo73*Δ (TC545) cells during PSM formation. (D) Localization of 3×GFPEnvy-Mss4 in wild-type (TNY375) cells during PSM extension (top) or after PSM closure (bottom), and *spo73*Δ (TNY376) cells during PSM formation. (E) Assessment of sporulation in wild-type *NDT80*-B/R P$_{CLB2}$-*degron-stt4* (TNY473) and producing 3×mKate2-Spo20$^{51-91}$-Sac1$^{2-517}$ from β-estradiol inducible promoter (TNY519). Wild-type *NDT80*-B/R (TNY168) was assessed as a control strain. More than 200 cells were observed in three independent colonies of each condition (for a total of > 600 cells). The bar graph shows the mean of the percentage of asci (N = 3). *, p < 0.05, ***, p < 0.001 (Tukey-Kramer test). (F) Left: Representative cells are shown. The perimeter of the PSM indicated by white arrow heads are shown. Right: Assessment of perimeter of the PSM in wild-type *NDT80*-B/R producing mKate2-Spo20$^{51-91}$ (TNY197), as a control, or wild-type *NDT80*-B/R P$_{CLB2}$-*degron-stt4* producing 3×mKate2-Spo20$^{51-91}$-Sac1$^{2-517}$ from β-estradiol inducible promoter (TNY519). More than 50 PSMs were measured in three independent colonies of each strain (for a total of 173 or 204 PSMs, respectively). The whiskers indicate the 10th and 90th percentiles and dots indicate outliers. ***, p < 0.001 (Student's t test). mK, mKate2. mKate2-Spo20$^{51-91}$, a PSM marker. Scale bar, 5 μm.

suggesting that PI(4,5)P$_2$ is synthesized in the PSM after closure (Fig 6C, bottom) [49]. Consistent with the localization of PI(4,5)P$_2$, Mss4, the unique PI4P 5-kinase in budding yeast, remained on the PM during PSM extension and appeared on the PSM after closure (Fig 6D). No significant differences were observed for FYVE$^{EEA1}$ and tandem PH$^{PLC\delta}$ localization in *spo73Δ* cells. However, 2×PH$^{PLC\delta}$ and Mss4 were not detected on the PSM even at later time points, which suggests a defect in PSM maturation. These results indicate that the growing PSM is distinctively abundant in PI4P in terms of phosphoinositides. The presence of PI4P on the PSM is functionally significant as lowering the PI4P level by simultaneous depletion of Stt4 and expression of 3×mKate2-Spo20$^{51-91}$-Sac1$^{2-517}$ in wild-type cells strongly compromised sporulation and PSM extension (Fig 6E and 6F). Thus, PI4P is important for sporulation and yet reduction of PI4P improves sporulation of *spo73Δ* mutants.

## Co-overexpression of Sac1$^{2-517}$ chimera encoding gene fusion and *VPS13* synergistically suppresses the defect of *spo73Δ* and *spo71Δ spo73Δ*

Spo73 and Spo71 may function as a complex to regulate Vps13 [23,37]. Therefore, we tested the ability of Sac1 targeted to the PSM to bypass sporulation defect of *spo71Δ* mutants. Indeed, overexpression of the Sac1$^{2-517}$ chimera protein restored some sporulation (6.2%) to a *spo71Δ* diploid and even to a *spo71Δ/spo71Δ spo73Δ/spo73Δ* (hereafter *spo71Δ spo73Δ*) double mutant (3.0%) (Fig 7A). In contrast, *vps13Δ* cells, even producing the Sac1$^{2-517}$ chimera, did not sporulate at all. Thus, lowered PI4P levels can partially bypass the need for these Vps13 regulatory proteins, though not for Vps13 itself, suggesting that Vps13 has a central role in the PSM extension and that the Spo71-Spo73 complex allows Vps13 to function at the PI4P-rich PSM. Overexpression of *VPS13* produced a very modest enhancement of sporulation in the *spo73Δ*, *spo71Δ*, and even in *spo71Δ spo73Δ* strain, but when both the Sac1$^{2-517}$ chimera encoding gene fusion and *VPS13* were overexpressed simultaneously the suppression was significantly enhanced (Figs 7B, S5A, and S5B). These results suggest that increasing the activity of Vps13 in the presence of lowered PI4P levels is the basis for bypass of *spo71Δ spo73Δ*. Enhanced suppression by overexpression of *VPS13* could also depend on Vps13 function in other localization than the PSM.

## Spo71 is important for Vps13 function beyond recruitment to the PSM

Previously, we showed that Spo71 was necessary to recruit Vps13 to the PSM [23]. However, that study was performed using a C-terminally tagged Vps13-GFP subsequently shown to be partially defective for function [25]. Therefore, the requirement for Spo71 in Vps13 localization to the PSM was confirmed using a functional form of Vps13 internally tagged with GFP-Envy [25,55]. Vps13^GFPEnvy localized to the PSM in wild-type and *spo73Δ* cells, but not in *spo71Δ* cells, confirming that Spo71 is required for PSM localization of Vps13 (Fig 7C). Residues 359–411 of Spo71 contain a "PxP" motif that binds to Vps13 VAB domain and fusion of this PxP-motif containing domain to an endosomal targeting sequence is sufficient to recruit Vps13 to the endosome in vegetative cells [26]. Consistent with these results, we found that presence of a chimeric protein composed of Spo71$^{359-411}$ and the mKate2-Spo20$^{51-91}$ could restore Vps13 localization to the PSM in the absence of full length Spo71 (Fig 7D). Importantly, although Spo71$^{359-411}$-mKate2-Spo20$^{51-91}$ restored Vps13 localization, it did not rescue the sporulation defect of *spo71Δ*. While *spo71Δ* cells overexpressing *VPS13* produced 20–50 colonies after ethanol treatment, those overproducing this chimera produced only a few colonies, which were similar to those carrying empty vector (S5C Fig). Thus, Spo71 must play a role in Vps13 function at the PSM beyond simply recruiting Vps13.

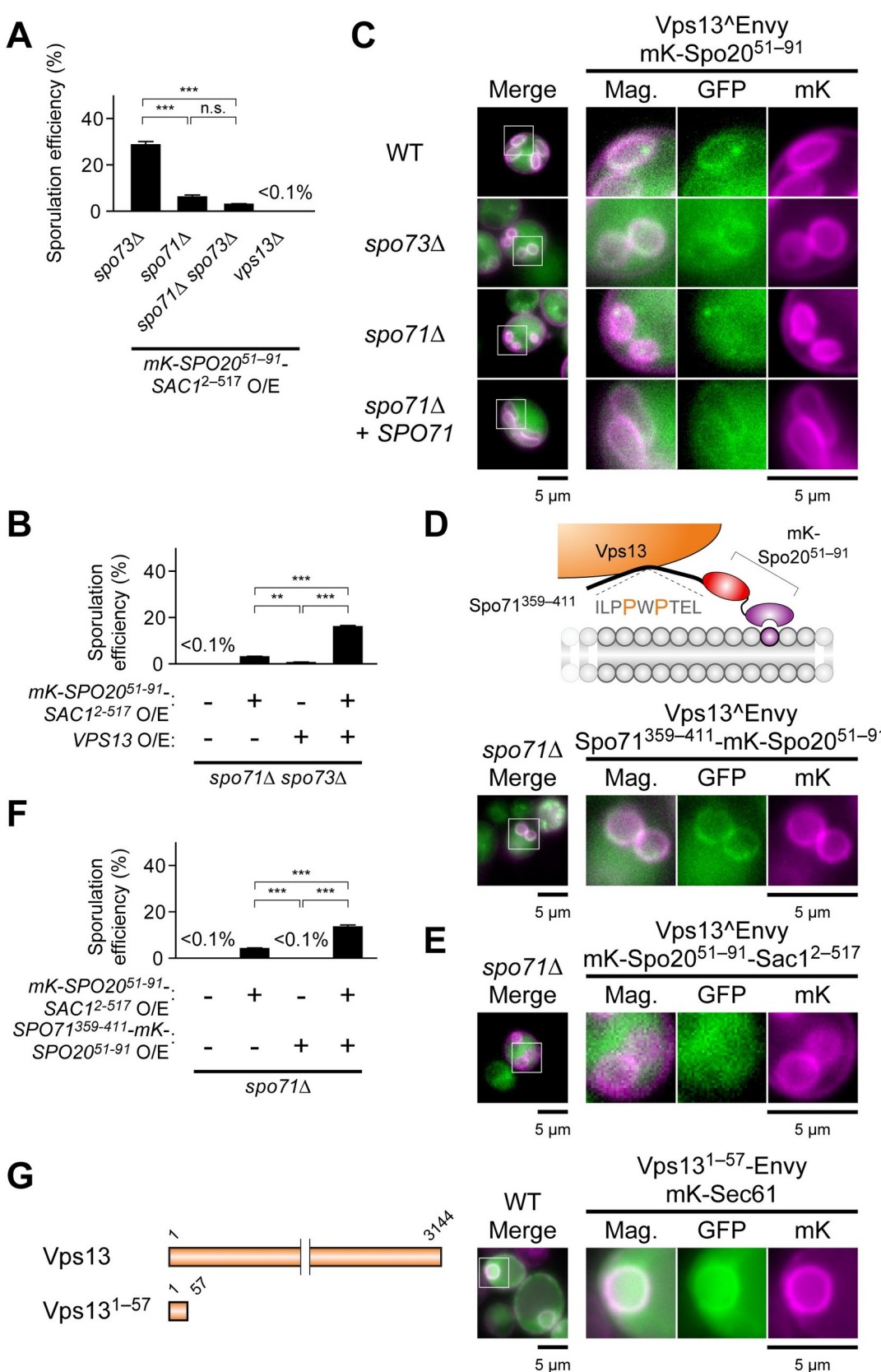

**Fig 7. Synergistic effects on suppression of Vps13 adaptor protein mutants.** (A) Assessment of sporulation in indicated mutants lacking Vps13 or the adaptor complex proteins overexpressing construct encoding mKate2-Spo20$^{51-91}$-Sac1$^{2-517}$. More than 200 cells were observed in three independent colonies of each strain (for a total of > 600 cells). The bar graph shows mean ± SEM of the sporulation efficiency (N = 3). These strains were *spo73Δ* (TC545), *spo71Δ* (TC581), *spo71Δ spo73Δ* (TNY637), and *vps13Δ* (TC572). ***, p < 0.001 (Tukey-Kramer test). (B) Assessment of sporulation in *spo71Δ spo73Δ* (TNY637) overexpressing construct encoding mKate2-Spo20$^{51-91}$-Sac1$^{2-517}$ and *VPS13*. More than 200 cells were observed in three independent colonies of each strain harboring indicated plasmids (for a total of > 600 cells). The bar graph shows mean ± SEM of the sporulation efficiency (N = 3). **, p < 0.01, ***, p < 0.001 (Tukey-Kramer test). (C) Localization of Vps13^GFPEnvy in wild-type (YFY83), *spo73Δ* (YFY85), and *spo71Δ* (YFY84) cells during PSM formation. Vps13^GFPEnvy localization was also observed in *spo71Δ* (YFY84) cells expressing *SPO71*. (D) Top: Diagram of Spo71$^{359-411}$-mKate2-Spo20$^{51-91}$. Bottom: Localization of Vps13^GFPEnvy in *spo71Δ* (YFY59) cells overexpressing construct encoding Spo71$^{359-411}$-mKate2-Spo20$^{51-91}$ during PSM formation. (E) Localization of Vps13^GFPEnvy in *spo71Δ* (YFY59) cells overexpressing construct encoding mKate2-Spo20$^{51-91}$-Sac1$^{2-517}$ during PSM formation. (F) Assessment of sporulation in *spo71Δ* (TC581) overexpressing construct encoding mKate2-Spo20$^{51-91}$-Sac1$^{2-517}$ and Spo71$^{359-411}$-mKate2-Spo20$^{51-91}$. More than 200 cells were observed in three independent colonies of each strain (for a total of > 600 cells). The bar graph shows mean ± SEM of the sporulation efficiency (N = 3). ***, p < 0.001 (Tukey-Kramer test). (G) Left: Diagram of Vps13$^{1-57}$. Right: Localization of Vps13$^{1-57}$-GFPEnvy in wild-type (AN120) cells expressing mKate2-Sec61 during vegetative growth. mKate2-Sec61, an ER marker. mK, mKate2. mKate2-Spo20$^{51-91}$, a PSM marker. Scale bar, 5 μm.

In contrast to Spo71$^{359-411}$-mKate2-Spo20$^{51-91}$, the presence of the Sac1$^{2-517}$ chimera could not restore the localization of Vps13 to the PSM, even when Vps13 was overproduced (Figs 7E and S5D). Moreover, the weak suppression of *spo71Δ* by Sac1$^{2-517}$ chimera was enhanced by the Spo71$^{359-411}$-mKate2-Spo20$^{51-91}$ as by overexpression of *VPS13* (Fig 7F). Thus, while Spo71 recruits Vps13 to the PSM, it must play an additional role in Vps13 function which may be related to PI4P.

## The Vps13 N-terminus can localize to the ER

Vps13 is proposed to act in lipid transfer between membranes, but what other organelle would be providing lipids to the PSM via Vps13 is not clear. The Atg2 protein is related to Vps13 and functions in transfer of lipids from the ER to the growing isolation membrane during autophagosome formation [30]. The N-terminal region of Atg2, which is homologous to Vps13, binds the ER and Atg2-Vps13 chimeras carrying N-terminal fragments of Vps13 complement *atg2Δ* [30,56], suggesting that the Vps13 N-terminus might have affinity for the ER. Indeed, the fusion protein composed of the extreme N-terminus of Vps13$^{1-57}$ and GFP colocalized with the ER marker protein Sec61 in vegetative yeast cells (Fig 7G). These results raise the possibility that Vps13 might simultaneously bind to the ER through its N-terminus and the PSM via Spo71 binding to its VAB domain and thereby link the two organelles.

## Tethers for the ER-PM contact sites localize along the PSM during sporulation, and are mislocalized in mutants lacking Vps13 or the adaptor complex proteins

The PSM is, in essence, a PM formed within the cell cytoplasm [31]. PI4P levels in the yeast PM are regulated in part by the Sac1 phosphatase through ER-PM contact sites. Six tethers, Ist2, Tcb1–3, Scs2, and Scs22, function redundantly to bridge the ER and the PM [14], and Ice2 has recently been reported to function as an additional tether protein [57]. At the ER-PM contact sites, Osh proteins are reported to exchange phosphatidylserine or sterol with PI4P between the ER and the PM [13]. The affinity of Vps13 to both the ER and the PSM suggests that ER-PSM contact sites are formed and, therefore, that ER-PM tethers might localize to these contact sites.

The localization of ER-PM tethers was examined in sporulating cells. Examining GFP fusions revealed that Ist2, Tcb3, and Tcb1 localize in close proximity to the PSM in wild-type cells, suggesting the existence of ER-PSM contact sites (Figs 8A and S6A). To confirm the

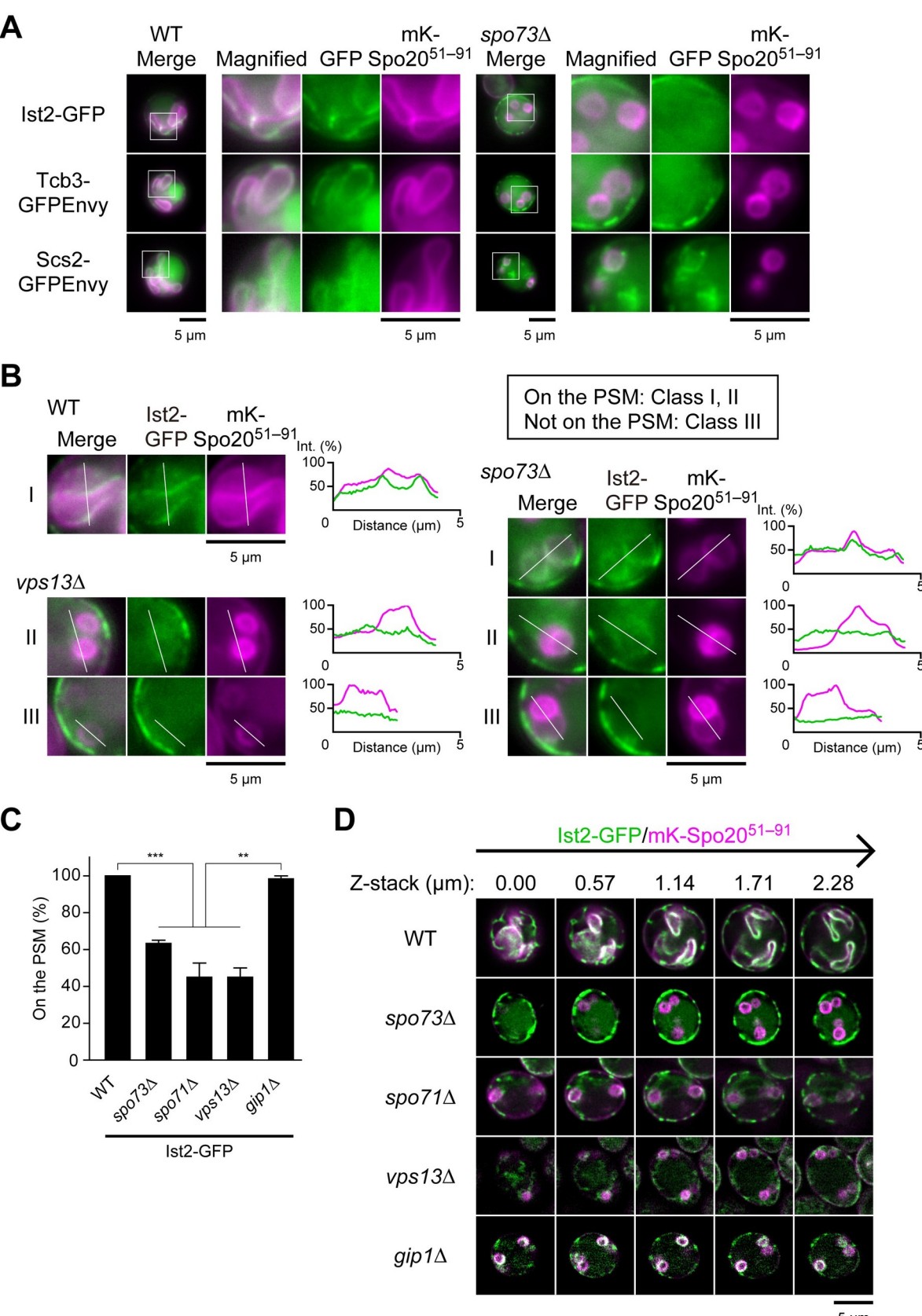

**Fig 8. ER-PM tethers localize along the PSM dependently on Vps13 and adaptor proteins.** (A) Localization of indicated tethers for ER-PM contact sites in wild-type (TNY375, left) and *spo73Δ* (TNY376, right) cells during PSM formation. (B) Localization of Ist2 in wild-type (TNY522), *spo73Δ* (TNY524), and *vps13Δ* (TNY545) cells during PSM formation. Classification in inset; On the PSM: Class I and II, Not on the PSM: Class III. Line plot profiles of the white line in each panel. (C) Assessment of localization of Ist2 in indicated cells during PSM formation. More than 20 cells were measured in three independent colonies of each strain (for a total of 60 cells, respectively). The bar graph shows mean ± SEM of the percentage of cells (N = 3). **, p < 0.01, ***, p < 0.001 (Tukey-Kramer test). (D) Z-stack images of the localization of Ist2-GFP in indicated cells during PSM formation. Images per 0.57 μm are shown. These strains were the same as in (C). mK, mKate2. mKate2-Spo20$^{51-91}$, a PSM marker. Scale bar, 5 μm.

presence of these contact sites, we used bimolecular fluorescence complementation (BiFC) and appended nonfluorescent GFP β1–10 ($β_{1-10}$) and GFP β11 strand ($β_{11}$) to the PSM marker mKate2-Spo20$^{51-91}$ and the ER transmembrane protein Tcb3, respectively [58]. In this experiment, simultaneous presence of mKate2-Spo20$^{51-91}$-$β_{1-10}$ and Tcb3-$β_{11}$ in wild-type cells resulted in GFP fluorescence along the PSM (S6B Fig), indicating the existence of ER-PSM contact sites.

In *spo73Δ* cells the localization of ER-PM tethers, including Ist2, to the PSM was disrupted (Figs 8A–8C, and S6A). Ist2 localization was also altered in *spo71Δ* and *vps13Δ* cells (Figs 8B, 8C, and S6C). About 50% of the cells showed no PSM localization (Class III: not on the PSM). To examine whether alteration of Ist2 localization is caused by PSM extension defect, we observed Ist2 localization in *gip1Δ/gip1Δ* (hereafter *gip1Δ*) cells, defective in PSM extension by a distinct mechanism. In contrast to *spo73Δ*, *spo71Δ* and *vps13Δ* cells, *gip1Δ* cells showed Ist2 localization similar to wild-type cells (Figs 8C and S6C, Class I: confined on the PSM). To more carefully examine whether Ist2-GFP signal overlaps with mKate2-Spo20$^{51-91}$ signal, we observed Ist2-GFP using super resolution microscopy in the different mutant strains. Strong Ist2-GFP signals were found on the PSM in wild-type and *gip1Δ* cells, but not in *spo73Δ*, *spo71Δ*, and *vps13Δ* cells (Fig 8D and S1–S5 Movies). Notably, in some mutant cells, Ist2 surrounded the PSM from the ascal cytoplasm (Figs 8B and S6C, Class II). To examine whether Ist2 signal is confined on the PSM, or expanded to the peri-PSM region, we measured PSM/peri-PSM ratio of Ist2 localization. In this analysis, we found that *spo73Δ*, *spo71Δ* and *vps13Δ* cells showed a low PSM/peri-PSM ratio compared to wild-type and *gip1Δ* cells, suggesting that precise localization of ER-PM tethers to the PSM is compromised in these mutants (S6D Fig). Notably, the localization of Ist2 was not restored by overproducing Sac1$^{2-517}$ chimera proteins in *spo73Δ* and *spo71Δ* cells (S6E Fig). Further, overproducing Spo71$^{359-411}$-mKate2-Spo20$^{51-91}$ also did not affect Ist2 localization (S6F Fig). These data suggest that suppression of the mutants is not related to restoring the localization of ER-PM tethers. As for Osh proteins, Osh2 and Osh3 localized on the PSM (S7 Fig), suggesting their involvement to PSM formation. Altogether, these results indicate that ER-PSM contact sites exist, and that Vps13 and the adaptor proteins recruit ER-PM tethers to these contact sites. Loss of this complex results in defects in the recruitment of ER-PM tethers, which could affect the PI4P levels in the PSM.

## Discussion

Vps13 localizes to various MCSs through binding to different adaptor proteins [26]. It is also known that various lipids contribute to proper Vps13 localization to different membranes [20,59,60]. Vps13 contains a hydrophobic cavity to transport lipids [17], thus, it has been suggested to be involved in efficient lipid transport between two organelles [28]. During sporulation, Vps13 is recruited to the PSM through its interaction with the Spo71-Spo73 adaptor complex. Our work showed that the regulation of PI4P levels at the PSM is necessary for Vps13 function, leading to PSM extension. Analysis of multicopy suppressors of *spo73Δ* revealed that the suppressors inhibit the PI4K complex on the PSM in a dominant-negative

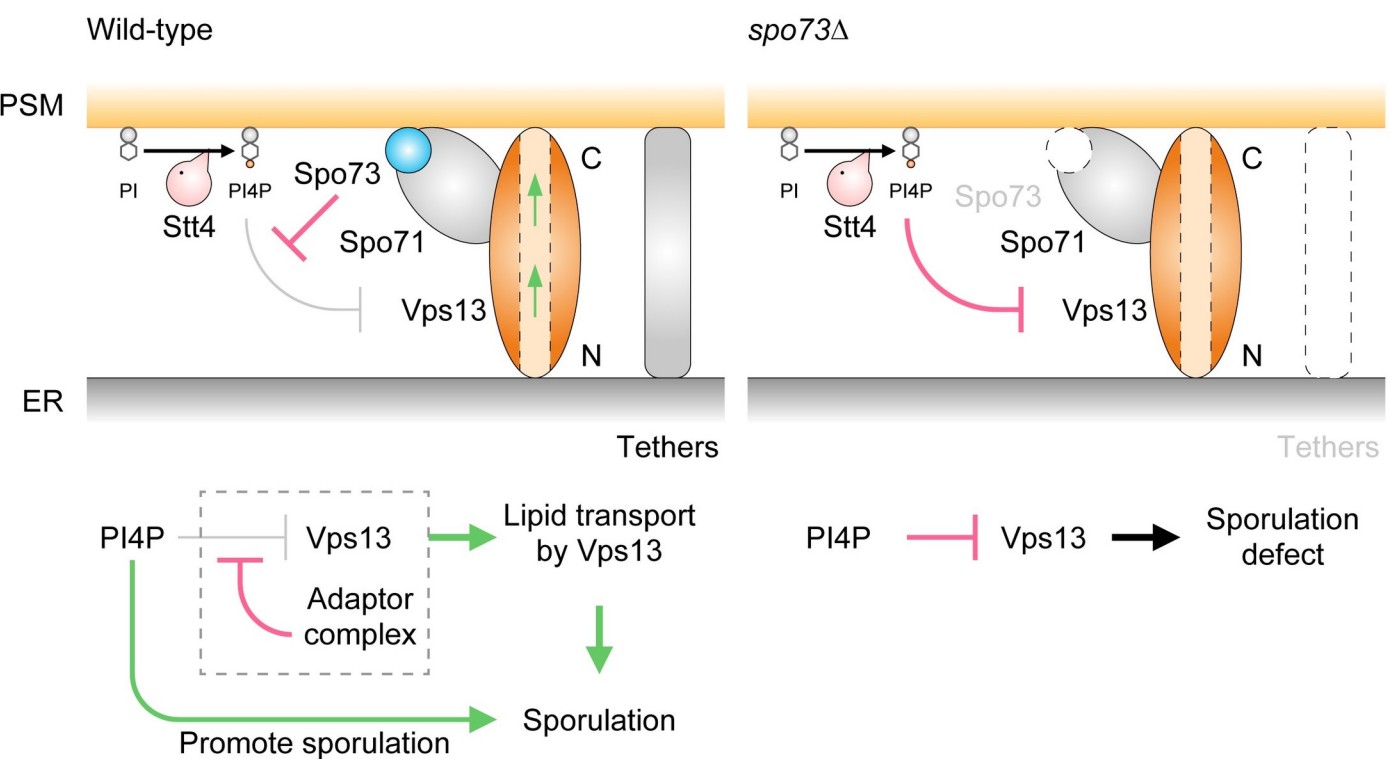

**Fig 9. Hypothetical model of this study.** Left: Model of PSM extension by Vps13, the adaptor complex, and ER-PM tethers through the regulation of PI4P levels in wild-type cells. Right: Model of PSM extension defect in *spo73Δ* cells.

manner. Downregulation of Stt4 at the onset of sporulation partially restored the defect of PSM extension in *spo73Δ*. Consistent with this, lowering PI4P levels in the PSM suppressed the defects of *spo73Δ*. Further, the overproduction of Vps13 and decreased levels of PI4P in the PSM synergistically suppressed *spo73Δ*, *spo71Δ*, and *spo71Δ spo73Δ* defects. Although recruitment of Vps13 to the PSM using PxP motif of Spo71 is not enough for suppression of *spo71Δ*, it can enhance suppression of *spo71Δ* by PSM-recruited Sac1$^{2-517}$ chimera, suggesting that PSM-recruited Vps13 is activated by lowering PI4P levels in the PSM. Considering that the *vps13Δ* defect cannot be suppressed by lowering PI4P levels in the PSM, we propose that Vps13 has a central role in transferring lipids to the PSM (Fig 9). This is consistent with a recent report on Vps13 suggesting a direct role in bulk lipid transfer [17]. Failure to provide sufficient lipids to promote membrane growth may be the source of the PSM extension defect in *vps13Δ* mutants.

## PSM is a PI4P enriched membrane compartment

Our analysis of different PIP-specific marker proteins shows that the expanding PSM is specifically enriched in PI4P. The localization of the Stt4 PI4K complex on the PSM suggests that it contributes directly to PI4P synthesis at the PSM. It has been suggested that an increase in the PI4P level in the PSM is required for PSM extension [54]. Consistent with this hypothesis, less fluorescence of a PI4P marker (GFP-1×PH$^{Osh2}$) was detected on the PSM in the *vps13Δ* and *spo71Δ* mutants than in wild-type cells [22]. However, in our previous work, we could not detect this difference in *spo73Δ* and *spo71Δ* mutants when only extending PSMs were observed [34]. In the present study, we successfully constructed a new marker to detect PI4P levels both on the Golgi and the PSM during sporulation. This marker allowed us to detect changes in the

PI4P levels in the PSM through a tug-of-war manner. Using this marker, we confirmed that the PI4P levels in the PSM decreased in cells producing a PI4P 4-phosphatase domain fused to a PSM marker protein, and that lowering the PI4P levels is sufficient for suppressing both the PSM extension and sporulation defects of *spo73Δ*.

It seems somewhat contradictory that the PSM is enriched in PI4P but that lowering the levels of this lipid can enhance sporulation in *spo73Δ*. However, lowering PI4P levels in wild-type cells by combining the downregulation of Stt4 and targeting the Sac1$^{2-517}$ chimera to the PSM drastically compromised sporulation, indicating that PI4P is required. These results suggest that a proper level of PI4P in the PSM is required for PSM extension.

It is also noteworthy that the PSM, though enriched in PI4P, appears to lack detectable PI(4,5)$P_2$. During PSM extension, Sfk1, a component of PI4K related to the regulation of PI(4,5)$P_2$ [61], did not localize on the extending PSM, and PI4P 5-kinase Mss4 appeared only on the mature PSM. In addition, we observed that PI(4,5)$P_2$ also appeared on the mature PSM, consistent with previous reports, which focused on the mature PSM [22,54]. Stage-specific regulation of PI(4,5)$P_2$ could also be important for PSM formation. PI(4,5)$P_2$ is suggested to be required for initial vesicle fusion [53], implying that there may be a low level of PI(4,5)$P_2$ at the PSM. Nonetheless, this difference in PIPs distinguishes the PSM from the PM and might be important for proper spore formation. For example, because PI(4,5)$P_2$ activates Rho1 and the cell wall integrity (CWI) pathway [41,62], it is possible that halting PI(4,5)$P_2$ synthesis on the growing PSM might inhibit aberrant spore wall formation. In higher eukaryotes, it has been reported that while depletion of PI(4,5)$P_2$ is maintained during cilia assembly, the synthesis of PI(4,5)$P_2$ triggers cilia decapitation [63]. Perhaps the balance of PI4P and PI(4,5)$P_2$ generally constitutes a signal involved in assembly and deformation of PM-like membranes.

## Vps13, adaptor complex and ER-PM tethers coordinately form ER-PSM contact sites

Our results reveal that the ER-PM tethers also localize to ER-PSM contact sites and that these ER-PSM contact sites are lost or reduced in *spo73Δ*, *spo71Δ*, or *vps13Δ* mutants. Why does localization of ER-PM tethers to the PSM require Vps13 and adaptor complex, while localization to the ER-PM contact sites do not require any protein? We have no evidence for direct interaction of Vps13 and adaptor complex with any of the tether proteins, but it is noteworthy that several of the tethers are thought to attached to the PM by binding to PI(4,5)$P_2$. Thus, the lowered amount of this lipid in the PSM may lead to a reliance on Vps13 and adaptor complex, either to directly recruit the tethers or perhaps just to bring the ER and PSM into close apposition. In either case, our results suggest an interplay between the two different MCS proteins. Vps13 and adaptor complex promote tether-mediated ER-PSM contact sites formation, and, in turn, ER-PM tethers stabilize ER-PSM contact sites and support lipid transfer by Vps13. This interplay may be a feature of Vps13, as a similar situation has been described at mitochondria-vacuole junctions where Vps13 appears to function in parallel to the vCLAMP contact site [64].

## A model for function of Vps13 and adaptor complex at the PSM

Altogether, we propose the following model for Vps13 and Spo71-Spo73 adaptor complex function at ER-PSM contact sites (Fig 9). In this model, Vps13 has a dual function, the establishment of contact sites and lipid transport. In wild-type cells, Vps13 and its adaptor complex establish a connection between the ER and the PSM that leads to the recruitment of ER-PM tethers to stabilize the ER-PSM contact sites. Vps13 functions in PSM extension by supporting bulk lipid transport from the ER. Although PSM extension requires PI4P which is synthesized

by Stt4, Vps13 is sensitive to PI4P in the PSM. This is alleviated by the existence of the adaptor complex which recruits, stabilizes, and may activate Vps13 on PI4P-rich membrane. In *spo73Δ* cells, loss of adaptor complex causes aberrant localization of the tethers, and ER-PSM contact sites may be destabilized or lost. Further, even though Spo71 can recruit Vps13 to the PSM in *spo73Δ* cells, Vps13 is inhibited by PI4P, resulting in PSM extension defect, which can be suppressed by a decrease in the PI4P levels in the PSM. In *spo71Δ* and *spo71Δ spo73Δ*, the defect is more severe, because loss of Spo71 compromises proper localization of Vps13 to the PSM. Besides lowering the PI4P levels in the PSM, theses mutants require Vps13 overproduction for efficient suppression. This implies that Vps13 has intrinsic affinity to both the ER and the PSM even without the adaptor complex. Multiple lipid binding regions have been identified across the Vps13 protein [20,59]. Considering recently reported structural analyses of Vps13 [17,27], our results suggest that N-terminal and C-terminal domains of Vps13 might have affinity for lipids of the ER and the PSM, respectively.

How does the Spo71-Spo73 adaptor complex alleviate the inhibitory effect of PI4P on Vps13 function? One possibility is that multiple PH domains of Spo71 might be involved in lipid binding for localization of Spo71, for regulation of the Vps13 activity, or both. Consistent with this possibility, co-overproduction of Sac1$^{2-517}$ chimera and Spo71$^{359-411}$ chimera enhanced suppression of *spo71Δ*. In this scenario, Spo73 binds Spo71 and could modulate the function of PH domain. For instance, a PH domain of Spo71 might mask PI4P and alleviate the inhibition of Vps13 function. Another possibility is that this complex could exert its effect through recruitment of ER-PM tethers and Osh proteins. At ER-PM contact sites, Osh proteins counter-transport PI4P from the PM to the ER where it is dephosphorylated by the Sac1 phosphatase. Loss of ER-PM contact sites results in the accumulation of PI4P on the PM [14,57]. Thus recruitment of ER-PM tethers and Osh proteins to ER-PSM contact sites by the adaptor complex might decrease the levels of PI4P in the PSM, which could alleviate the inhibition of Vps13 function.

In our model, PI4P plays a key role in regulating Vps13 function. An alternative possibility is suggested by recent reports demonstrating that PI levels in the PM of mammalian cells are actually low relative to the levels of PI phosphates [65,66]. If this is true in the PSM, it is possible that interfering with Stt4 causes suppression not by lowering PI4P levels but by elevating PI levels. In this case, PI in the PSM would boost Vps13 transfer activity. Determining whether the effects on Vps13 are caused by either PI4P or PI will require direct measurement of PSM PI and PI4P levels.

## A conserved strategy for de novo membrane expansion

The PSM superficially resembles formation of an autophagosome in that it is a membrane compartment formed de novo within the cytoplasm that expands and extends to engulf and, ultimately, enclose cytoplasmic contents within a double membrane [31,67]. Our study reveals a potential functional parallel between the two processes. Autophagy begins with the fusion of vesicles at or on a proteinaceous structure termed the pre-autophagosomal structure (PAS) to form an isolation membrane. Expansion of the isolation membrane into an autophagosome requires its continued supply of membrane lipids mediated both by vesicle fusion and by bulk lipid transfer from the ER through the Vps13-related Atg2 protein [30,68]. Similarly, PSM formation begins with the fusion of post-Golgi vesicles on the surface of a proteinaceous matrix, in this case on the spindle pole body (SPB). Expansion of the PSM requires continued delivery of vesicles from the Golgi and, our present study suggests, also bulk lipid transport from the ER mediated by Vps13. The specific proteins involved in the two processes are different. Atg2 and Vps13 and their partners are different complexes, the PAS and SPB are distinct structures,

and the vesicles that coalesce on the two structures are distinct. Nonetheless these results suggest that the use of bulk lipid transfer proteins of the Vps13/Atg2 type to rapidly expand a novel intracellular membrane compartment might be a common strategy in eukaryotic cells.

## Materials and methods

### Yeast strains and media

Standard media and genetic techniques were used unless otherwise noted [69]. Yeast strains used are listed in S1 Table. In brief, linearization of integration vectors was carried out with following enzymes. TNP1139 and TNP1546 were digested with BssHII. TNP0351 and TNP0352 were digested with EcoRV. TNP1237, TNP1539, and TNP1828 were digested with PstI. TNP1547 was digested with SmaI. TNP1059 and TNP0210 were digested with StuI. TNP1622 was digested with XbaI. PCR for genetic modification of the yeast genome was carried out with following oligonucleotides and templates: *hphNT1*::P$_{4×lexA}$-*9×Myc-NDT80* (TN334, TN335, TNP1065), *natNT2*::P$_{CLB2}$-*degron-stt4* (TN193, TN265, TNP1119), *spo73*::*kanMX6* (IC8, IC9, genome of TC545), *spo71*::*kanMX6* (HT403, HT404, genome of TC581), *vps13*::*kanMX6* (TN433, TN434, genome of TC572), *gip1*::*kanMX6* (HT5, TN391, genome of TC544), *TCB2*::*GFPEnvy*::*HIS3MX6* (TN872, TN873, TNP1034), *spo71*::*natNT2* (TN828, TN829, TNP1923), *vps13^TRP1* (TN631, TN632, TNP1002), *VPS13-GFPEnvy(1360)-loxP-HIS3MX6-loxP* (TN831, TN832, TNP1931).

YFY39 (*VPS13^GFPEnvy MAT*α) and YFY40 (*VPS13^GFPEnvy MAT***a**) were constructed as follows: First, as a healing fragment, *VPS13-GFPEnvy$_{1360}$-frag* was amplified with TN419, TN634, and TNP0532. Next, YFY30 and YFY29 cells harboring *VPS13-GFPEnvy(1360)-loxP-HIS3MX6-loxP* were transformed with pRS426-Cas9-SkHIS3-381 (TNP1956). These cells constitutively expressed Cas9 from the *TEF1* promotor. Next, these cells were cultured in YPD medium and incubated with healing fragments, generating the *VPS13-GFPEnvy$_{1360}$* strain. Finally, TNP1956 was kicked out by inoculating these strains on 5-fluoroorotic acid (5-FOA)-containing medium.

### Plasmids

The plasmids used are listed in S2 Table, and oligonucleotides used are listed in S3 Table. The *mTagBFP2* and *superfolderGFP* fragments were synthesized by gBlocks (Integrated DNA Technologies (IDT), Coralville, Iowa).

pRS314-VPS13-GFPEnvy(1360) (TNP0532) was constructed as follows: First, pRS424-VPS13-TRP1(1360)-frag (TNPP110) was constructed with TN635, TN636, the genome of TNY475 as a template, and pRS424 as a vector (digested with SacII and KpnI). Next, pRS314-P$_{TEF1}$-N-GFPEnvy-T$_{CYC1}$ (TNP1355) and TNPP110 were digested with NotI and BamHI, and the *GFPEnvy* fragment was inserted into TNPP110, generating pRS424-VPS13-GFPEnvy(1360)-frag (TNPP109). Next, pRS426-VPS13 (TNP0449) and pRS314 were digested with NotI and KpnI, and the *VPS13* fragment was inserted into pRS314, generating pRS314-VPS13 (TNP0448). Next, the *VPS13*-3104-3192bp fragment was amplified with TN701, TN702, and the genome of AN120 as a template, and the *VPS13-GFPEnvy(1360)* fragment was amplified with TN633, TN634, and TNPP109 as a template, and the *VPS13*-4163-5484bp fragment was amplified with TN703, TN704, and the AN120 genome as a template. To generate TNP0532, TNP0448 was digested with BamHI, cleaving at two BamHI sites in *VPS13*, and subjected to an In-fusion reaction (Clontech) with three *VPS13* fragments.

pFA6a-VPS13-GFPEnvy(1360)-loxP-HIS3MX6-loxP-Long (TNP1931) was constructed as follows: First, the *VPS13-GFPEnvy(1360)-frag-Long* fragment was amplified with TN801, TN802, and TNP0532 as a template, and subjected to an In-fusion reaction with HindIII/SpeI-

digested pFA6a-kanMX6, generating pFA6a-VPS13-GFPEnvy(1360)frag-Long (TNP1930). Next, the *loxP-HIS3MX6-loxP* fragment was amplified with TN807, TN810, and pFA6a-loxP-HIS3MX6-loxP (TNP1016) as a template, and subjected to an In-fusion reaction with BamHI-digested TNP1016, generating TNP1931.

## Sporulation and the *NDT80*-block/release system

Sporulation was performed as previously described [33,35]. Synchronous sporulation in the estradiol-inducible *NDT80*-block/release system was carried out as previously described [70]. At 6 h postinduction, β-estradiol was added to 2 μM to induce expression of *NDT80*.

## Microscopy

Differential interference contrast (DIC) images and fluorescence images were obtained with a BX71 microscope (Olympus, Tokyo, Japan), a Quantix 1400 camera (Photometrics), and IPLab 3.7 software (Scanalytics). In observation of Osh2-P4M, images were obtained with a BZ-X710 microscope (Keyence, Osaka, Japan), and processed with Fiji (ImageJ; National Institutes of Health, http://rsb.info.nih.gov/ij/). PSM perimeter measurements were performed as previously described [34]. PSM perimeters were measured by Fiji. Confocal images were obtained with a ZEISS LSM 880 with Airyscan (Carl Zeiss, Oberkochen, Germany) with ZEISS Efficient Navigation (ZEN) software (Carl Zeiss).

## Screening of multicopy suppressors of *spo73Δ*

Screening of multicopy suppressors was performed as previously described [71]. *spo73Δ* cells were transformed with a genomic library constructed using TC544 (*gip1Δ*) genomic DNA, sporulated at 34˚C, and clones that formed ethanol-resistant spores were isolated. A total of $1.3 \times 10^4$ colonies was subjected to screening, and three plasmids were isolated as multicopy suppressors.

## Antibodies

The following antibodies were used: mouse anti-mini-AID (1:1000; M214-7; Medical & Biological Laboratories (MBL), Aichi, Japan), mouse anti-Pgk1 (1:500, 459250, Thermo Fisher Scientific, Waltham, MA, United States), anti-mouse IgG (whole molecule) (1:5000; A9044, Sigma-Aldrich, St. Louis, MO, United States).

## Western blot analysis

Western blot analysis was performed as previously described [35]. In brief, the YPD, YPA, and sporulating cultures were treated with trichloroacetic acid (TCA) (final concentration: 6%), resuspended in urea buffer (6 M urea, 50 mM Tris-HCl [pH 7.5], 5 mM EDTA, 1% SDS, 1 mM phenylmethylsulfonyl fluoride), and vortexed with a multibead shocker to disrupt cells. These samples were boiled at 95˚C for 10 min, mixed with 3×SDS sample buffer (150 mM Tris-HCl [pH 6.8], 6% SDS, 18% β-mercaptoethanol, 30% glycerol, 0.3 mg/ml bromophenol blue), and subjected into SDS-PAGE. Proteins separated on SDS-PAGE gels were transferred to polyvinylidene difluoride membranes. Membranes were blocked in 5% skim milk in TBS/T (Tris-buffered saline, 50 mM Tris-HCl [pH 7.5], 150 mM NaCl, 0.1% Tween-20) for 1 h. Membranes were probed for 1 h with the primary antibodies, washed three times with TBS/T for 10 min, and treated with the horseradish peroxidase-conjugated secondary antibody for 1 h. After being washed three times with TBS/T for 10 min, membranes were treated with

ImmunoStar LD (296–69901; Wako, Osaka, Japan), and images were obtained with an Image-Quant LAS-4000mini (GE Healthcare, Little Chalfont, UK).

## Quantification of Ist2 localization

Quantification of Ist2 localization was performed by ImageJ as follows: First, the image of mKate2-Spo20$^{51-91}$ was processed by binarizing the mKate2 channel and defining the regions of PM and cytoplasm, or by manually defining the region of PSM, respectively. In defining the region of Peri-PSM, the region 4 px larger than the PSM region was selected, the PSM region was subtracted from the larger region, and the residual surrounding region was designated as the Peri-PSM region. Next, the image of Ist2-GFP was masked by each region, the intensity of Ist2-GFP in each region was integrated, and PSM/peri-PSM ratios were calculated.

## Movies

S1–S5 Movies. Z-stack images of the localization of Ist2-GFP in indicated cells during PSM formation. Wild-type (S1 Movie, TNY522), *spo73Δ* (S2 Movie, TNY524), *spo71Δ* (S3 Movie, TNY544), *vps13Δ* (S4 Movie, TNY545), and *gip1Δ* (S5 Movie, TNY546) cells were observed with confocal microscopy. PSM, mKate2-Spo20$^{51-91}$, a PSM marker. Scale bar, 5 μm.

## Supporting information

**S1 Fig. The PI4K complex localizes on the PSM, and suppression occurs through a dominant-negative effect.** (A) Assessment of Stt4 activity. *stt4-4* (AAY102) was transformed indicated plasmid, and grown on SD plates for 2 days at permissive (30˚C) or nonpermissive (37˚C) temperature. (B) Assessment of sporulation in *spo73Δ* (TC545) overexpressing wild-type *STT4* (WT) or *STT4-KD*. More than 200 cells were observed in three independent colonies of each strain harboring indicated plasmids (for a total of > 600 cells). The bar graph shows the mean of the percentage of asci (N = 3). (C and D) Localization of GFPEnvy-Ypp1 (C) and Sfk1-GFP (D) in wild-type (AN120) or *spo73Δ* (TC545) cells during PSM formation. mK, mKate2. mKate2-Spo20$^{51-91}$, a PSM marker. Scale bar, 5 μm.
(PDF)

**S2 Fig. Selective depletion of PI4P in the PSM suppresses the defects of *spo73Δ*.** Localization of Sac1$^{2-517}$ chimera proteins in *spo73Δ* (TC545) cells during PSM formation. mR, mRFP. PD, phosphatase-dead. Scale bar, 5 μm.
(PDF)

**S3 Fig. Changes in PI4P levels in the PSM can not be detected using conventional PI4P biomarkers.** (A) Localization of GFP-1×PH$^{Osh2}$ (left) or GFP-2×PH$^{Osh2}$ (right) in *spo73Δ* (TC545) cells overproducing mKate2-Spo20$^{51-91}$-Sac1$^{2-517}$ during PSM formation. (B) Localization of GFP-P4M in wild-type cells producing Sec7-mRFP (TNY643) during vegetative growth. Sec7-mRFP, a Golgi marker. (C) Localization of GFP-P4M in wild-type (AN120) and *spo73Δ* (TC545) cells during PSM formation. mKate2-Spo20$^{51-91}$, a PSM marker. (D) Localization of GFP-P4M in wild-type cells producing Sec7-mRFP and mTagBFP2-Spo20$^{51-91}$ (TNY643) during PSM formation. Sec7, a Golgi marker. mTagBFP2-Spo20$^{51-91}$, a PSM marker. mK, mKate2. Scale bar, 5 μm.
(PDF)

**S4 Fig. Changes in PI4P levels in the PSM can be detected using an improved PI4P biomarker.** (A and B) Localization of GFP-2×PH$^{Osh2}$ (A), or GFP-Osh2-P4M (B) in wild-type (AN120) cells during vegetative growth. Two representative images are shown for each strain.

(C) Localization of indicated PI4P markers in *stt4-4* (AAY102) or *pik1-83* (AAY104) cells during vegetative growth. These cells were observed after incubation at indicated temperatures for 60 min. (D) Localization of GFP-Osh2-P4M in wild-type cells expressing Sec7-mRFP and mTagBFP2-Spo20$^{51-91}$ (TNY642) during PSM formation. Sec7-mRFP, a Golgi marker. mTagBFP2-Spo20$^{51-91}$, a PSM marker. Scale bar, 5 μm.
(PDF)

**S5 Fig. Synergistic effects on suppression of Vps13 adaptor protein mutants.** (A and B) Assessment of sporulation in *spo73Δ* (TC545, A) *spo71Δ* (TC581, B) overexpressing construct encoding mKate2-Spo20$^{51-91}$-Sac1$^{2-517}$ and *VPS13*. More than 200 cells were observed in three independent colonies of each strain harboring indicated plasmids (for a total of > 600 cells). The bar graph shows mean ± SEM of the sporulation efficiency (N = 3). **, p < 0.01, ***, p < 0.001 (Tukey-Kramer test). (C) Ethanol resistance assay. *spo71Δ* (TC581) cells overexpressing *VPS13* or construct encoding Spo71$^{359-411}$-mKate2-Spo20$^{51-91}$, or harboring empty vector, were sporulated for 2 days. Samples of $1 \times 10^6$ cells of each transformant were treated with or without 26% ethanol (+EtOH or -EtOH) and inoculated onto YPD plates for 2 days. Magnification of inverted images of +EtOH YPD plates are shown. (D) Localization of overproduced Vps13^GFPEnvy in wild-type (AN120), *spo73Δ* (TC545) and *spo71Δ* (TC581) cells overexpressing construct encoding mKate2-Spo20$^{51-91}$-Sac1$^{2-517}$ during PSM formation. mK, mKate2. Scale bar, 5 μm.
(PDF)

**S6 Fig. ER-PM tethers localize along the PSM dependently on Vps13 and adaptor proteins.** (A) Localization of indicated tethers for ER-PM contact sites in wild-type (TNY375, left) and *spo73Δ* (TNY376, right) cells during PSM formation. (B) Assessment of proximity of the PSM and the ER by BiFC assay. Wild-type (TNY375) cells producing mKate2-Spo20$^{51-91}$-β$_{1-10}$, Tcb3-GFP-β$_{11}$, and mKate2-Spo20$^{51-91}$ were observed during PSM formation. (C) Localization of Ist2 in *spo71Δ* (TNY544) and *gip1Δ* (TNY546) cells during PSM formation. Classification in inset; On the PSM: Class I and II, Not on the PSM: Class III. Line plot profiles of the white line in each panel. (D) Assessment of localization of Ist2 in indicated cells during PSM formation. Each strain expressing Ist2-GFP and mKate2-Spo20$^{51-91}$ was observed during PSM formation, and PSM/peri-PSM ratios of the fluorescence of Ist2-GFP were calculated. More than 20 cells were measured in three independent colonies of each strain (for a total of 60 cells, respectively). The bar graph shows mean ± SEM of the percentage of cells (N = 3). *, p < 0.05, **, p < 0.01, ***, p < 0.001 (Tukey-Kramer test). (E) Assessment of localization of Ist2 in *spo73Δ* (TC611) and *spo71Δ* (TC609) cells overexpressing constructs encoding phosphatase active (WT) or phosphatase-dead (PD) Sac1$^{2-517}$ chimera proteins during PSM formation. More than 20 cells were measured in three independent colonies of each strain (for a total of 60 cells, respectively). The bar graph shows mean ± SEM of the percentage of cells (N = 3). n.s., not significant (Tukey-Kramer test). (F) Assessment of localization of Ist2 in *spo71Δ* (TC609) cells overexpressing constructs encoding Spo71$^{359-411}$-mKate2-Spo20$^{51-91}$ during PSM formation. More than 20 cells were measured in three independent colonies of each strain (for a total of 60 cells, respectively). The bar graph shows mean ± SEM of the percentage of cells (N = 3). n.s., not significant (Student's t test). mK, mKate2. mKate2-Spo20$^{51-91}$, a PSM marker. Scale bar, 5 μm.
(PDF)

**S7 Fig. Localization of Osh proteins during PSM formation.** Localization of indicated Osh proteins in wild-type (TNY375) cells during PSM formation. mKate2-Spo20$^{51-91}$, a PSM

marker. Scale bar, 5 μm.
(PDF)

**S1 Movie. Z-stack image of the localization of Ist2-GFP in wild-type cell.**
(MP4)

**S2 Movie. Z-stack image of the localization of Ist2-GFP in *spo73Δ* cell.**
(MP4)

**S3 Movie. Z-stack image of the localization of Ist2-GFP in *spo71Δ* cell.**
(MP4)

**S4 Movie. Z-stack image of the localization of Ist2-GFP in *vps13Δ* cell.**
(MP4)

**S5 Movie. Z-stack image of the localization of Ist2-GFP in *gip1Δ* cell.**
(MP4)

**S1 Table. Strains used for this study.**
(PDF)

**S2 Table. Plasmids used for this study.**
(PDF)

**S3 Table. Oligonucleotides used for this study.**
(PDF)

## Acknowledgments

We are grateful to T. Maeda (Hamamatsu Univ. School of Medicine), M. Onishi (Duke Univ.), T. Noda (Osaka Univ.), Addgene, and the National Bio-Resource Project (NBRP) for reagents. We are also grateful to T. Suda for supporting analysis of micrographs, T. Maeda, N. Fujita (Tokyo Institute of Technology), and members of the Touhara Lab for helpful advice.

## Author Contributions

**Conceptualization:** Tsuyoshi S. Nakamura, Aaron M. Neiman, Hiroyuki Tachikawa.

**Formal analysis:** Tsuyoshi S. Nakamura, Yasushi Okada, Hiroyuki Tachikawa.

**Funding acquisition:** Tsuyoshi S. Nakamura, Aaron M. Neiman, Hiroyuki Tachikawa.

**Investigation:** Tsuyoshi S. Nakamura, Yasuyuki Suda, Kenji Muneshige, Yuji Fujieda, Yuuya Okumura, Ichiro Inoue, Takayuki Tanaka, Tetsuo Takahashi, Hideki Nakanishi.

**Methodology:** Tsuyoshi S. Nakamura, Yasushi Okada.

**Project administration:** Hiroyuki Tachikawa.

**Resources:** Hideki Nakanishi, Xiao-Dong Gao, Aaron M. Neiman.

**Supervision:** Hiroyuki Tachikawa.

**Validation:** Yasuyuki Suda, Ichiro Inoue, Hiroyuki Tachikawa.

**Writing – original draft:** Tsuyoshi S. Nakamura, Hiroyuki Tachikawa.

**Writing – review & editing:** Tsuyoshi S. Nakamura, Yasuyuki Suda, Aaron M. Neiman, Hiroyuki Tachikawa.

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
