## [Decision Letter · Decision Letter 0]

8 Apr 2021

Dear Dr TACHIKAWA,

Thank you very much for submitting your Research Article entitled 'Suppression of Vps13 adaptor protein mutants reveals a role for PI4P in regulating Vps13 function' to PLOS Genetics.

The manuscript was fully evaluated at the editorial level and by independent peer reviewers. The reviewers appreciated the attention to an important problem, but raised some substantial concerns about the current manuscript. Based on the reviews, we will not be able to accept this version of the manuscript, but we would be willing to review a much-revised version. We cannot, of course, promise publication at that time.

If you decide to revise the manuscript for further consideration at PLOS Genetics, please aim to resubmit within the next 60 days, unless it will take extra time to address the concerns of the reviewers, in which case we would appreciate an expected resubmission date by email to plosgenetics@plos.org.

[LINK]

We are sorry that we cannot be more positive about your manuscript at this stage. Please do not hesitate to contact us if you have any concerns or questions.

Yours sincerely,

Teresa Zoladek

Guest Editor

PLOS Genetics

Gregory P. Copenhaver

Editor-in-Chief

PLOS Genetics

Reviewer's Responses to Questions

**Comments to the Authors:**

Reviewer #1: Review is attached as a word document.

Reviewer #2: This is very well written and executed paper, only minor corretions needed, it was a pleasure to read.

Reviewer #3: Nakamura et al. used genetic approach accompanied by a wide set of cytological tools to study the role of the conserved lipid transfer protein Vps13 in formation of the prospore membrane (PSM) during sporulation in budding yeast. The paper presents several novel findings that are well supported by the experimental data. However, in my opinion, the major conclusion of the paper that phosphatidylinositol-4-phosphate (PI4P) levels regulate activity of Vps13 at the PSM seems a little far-fetched. There is a clear interplay between the PI4P content in the PSM and the function of Vps13 complex; however, the mechanism of Vps13 regulation by PI4P has not been elucidated mechanistically and the influence of PI4P on Vps13 may also be indirect. In the discussion, the authors themselves proposed an alternative interpretation of their results pointing to the possible involvement of PI levels instead of PI4P in Vps13 regulation (page 13, lines 431-437).

The experimental work is generally well-done and described. However, several issues need to be addressed or clarified before acceptance. I also suggest some changes in figure layouts and abbreviations to improve the perception of content.

Major points:

The title does not reflect well major findings of the paper. The title should rather emphasize the role of Vps13 in the establishment of the ER-PSM contact sites and demonstration of high PI4P levels in the PSM, both of which are essential for extension and maturation of the PSM.

Is there any effect of STT4frag, STT4, STT4-KD and EFR3 overexpression on sporulation efficiency in wild-type background?

Overexpression of STT4-KD in the spo73 mutant resulted in 4.3% sporulation efficiency, whereas spo73 cells overexpressing STT4frag showed 4 times higher sporulation rate. Both Stt4 variants lack kinase activity and should show similar levels of suppression. Please comment this discrepancy.

Figure 1C: it is shown that overexpression of 651-782 fragment of Efr3 in the spo73 mutant resulted in 2-fold higher sporulation efficiency compared to full length Efr3. Please comment this discrepancy.

Page 6, line 183: it is stated that “the downregulation of Stt4 directly suppressed the defect of spo73del (Fig. 3 D)”. Can suppression, as a genetic term, be direct or indirect?

Figures 4D and 6F lack representative photos showing different perimeters of PSM in indicated strains.

Page 8, lines 241-243, 248-249 and Figure 6B: what was the purpose of checking PI3P distribution in spo73 vs wild-type cells?

Figure 7B: what is the effect of Vps13 and K20-Sac1-P on sporulation efficiency in spo71 and spo73 single mutants? Also, what is the localization pattern of Vps13 when overexpressed in spo mutants? Does it show any colocalization with PSM?

Figure layouts:

Figure 1A, 2A, 3B, 3D, 4C and 6E: first, is it necessary to show percentage of all ascus classes observed? The authors never referred to these classes in the text. In my opinion, showing just the total percentage of asci having at least one spore is sufficient. Second, I suggest graphs showing sporulation efficiency (%) in the way presented in Figures 7A and 7B and the maximum value of Y axis does not need to be 100% (same for Fig. 7A and 7B). This would help to read the % values in Figures without referring to the text.

I suggest to combine Figure 2A and Figure S1A into a separate supplemental figure.

I suggest to incorporate panels Fig.S1B and Fig.S1E into panels Fig.2B and Fig.2C, respectively. The reader of the present version has to jump back and forth between Fig.2 and Fig.S1.

Figure 4: I suggest to present panel B as a supplemental figure.

Abbreviations:

Some abbreviations of chimeric proteins are confusing and I suggest to stick to full versions both in the text and Figures for nomenclature consistency and overall clarity of the content.

- “Sac1-P” may also refer to phosphorylated form of the protein; I suggest using “Sac1<2-517>superscript”

- “R20” may also refer to Arg20; I suggest to use always the full name “mRFP-Spo20<51-91>superscript”

- Use 3xmKate2-Spo20<51-91>-Sac1<2-517> instead of “3xK20-Sac1-P” as “K20” may also refer to Lys20 and such abbreviation may indicate three copies of both mKate2 and Spo20.

- Use “Spo71<359-411>superscript” instead of “PxP-K20”.

Minor points:

Page 3, line 92: correct citation style.

Page 5, line 129: provide rationale behind isolation of ethanol-resistant spores.

Legend of Figure 1E: Stt4(136-814) and Efr3(651-782) were used a control. A control of what? Please explain in the legend.

Legend of Figure S2, panel B: localization of GFP-P4M is shown during vegetative growth not during PSM formation as described in the legend.

Legend of Figure S2, panel C: localization of GFP-P4M is shown during PSM formation not during vegetative growth as described in the legend.

Please describe the mRFP-Spo2051-91 fusion protein when first mentioned on page 5, line 159. First description is on page 6, lines 193-194.

Please explain in the text what are Sfp1 (page 5, line 159), pik1 ts mutant (page 7, line 221), gip1del (page 10, line 323) and why they were tested.

Page 7, lines 219-221: instead of saying that “localization was altered”, please describe in detail localization changes of Osh2-P4M in stt4-4 and pik1-83 cells (Figure S3C).

Figure 9 should be titled “Hypothetical model..”.

**Have all data underlying the figures and results presented in the manuscript been provided?**

Reviewer #1: Yes

Reviewer #2: Yes

Reviewer #3: Yes

PLOS authors have the option to publish the peer review history of their article (what does this mean?). If published, this will include your full peer review and any attached files.

Reviewer #1: No

Reviewer #2: No

Reviewer #3: No

---

## [Editor Report · Decision Letter 1]

17 Jul 2021

Dear Dr TACHIKAWA,

Thank you very much for submitting your revised Research Article entitled 'Suppression of Vps13 adaptor protein mutants reveals a central role for PI4P in regulating prospore membrane extension' to PLOS Genetics.

The manuscript was fully evaluated at the editorial level and by reviewer 1. The reviewers appreciated the attention to an important topic but identified some concerns that we ask you address in a revised manuscript

We therefore ask you to modify the manuscript according to the review recommendations. Your revisions should address the specific points made by each reviewer.

We hope to receive your revised manuscript within the next 7 days. If you anticipate any delay in its return, we would ask you to let us know the expected resubmission date by email to plosgenetics@plos.org.

[LINK]

Yours sincerely,

Teresa Zoladek

Guest Editor

PLOS Genetics

Gregory Copenhaver

Editor-in-Chief

PLOS Genetics

The manuscript is now greatly improved, according to reviewers suggestions.

However, we suggest minor corrections which are listed below.

The authors added in the discussion that Vps13 may be recruited to membranes due to interactions with lipids, in addition to interactions with proteins, but this information should also be included on line 57.

The conclusion in P10,L306 should be rather extended to point the other possibility: “and enhanced suppression by overexpression of VPS13 may depend on Vps13 function in other localisation than PSM.”

Figure 7

Inconsistency in panels, in C Vps13 ^ Envy and mK-Spo20 51-91, but in D,E and G fusion proteins are listed above the line and pictures are labelled GFP and mK.

To standardize, mKate in panel G should be shortened to mK, as in the rest of the text. It is not known from the figure which protein tagged with GFP is shown in G.

“Construct” is a laboratory language. Better use terms: plasmid, gene, gene fusion, instead (L239, 280).

P4, L127 Overexpression of EFR3 or an fragment of STT4 gene

P5, L159 we constructed a plasmid encoding a kinase-dead (KD) Stt4…. based on the mutant Pik1 (or pik1 mutant)

P7, L203, we constructed genes encoding fused

L2-4, PSM, or Dtr1.

P9, L269 chimera encoding gene fusion

P14, L443, Vps13 overproduction

L451, co-overproduction

P28, L845 Overexpression of EFR3 or truncated STT4

P29, L88, add “The” to avoid starting the sentence with small letter.

L30, L916 markers used

P31, L936 Wild-type

---

## [Editor Report · Decision Letter 2]

20 Jul 2021

Dear Dr HIROYUKI TACHIKAWA,

We are pleased to inform you that your manuscript entitled "Suppression of Vps13 adaptor protein mutants reveals a central role for PI4P in regulating prospore membrane extension" has been editorially accepted for publication in PLOS Genetics. Congratulations!

Yours sincerely,

Teresa Zoladek

Guest Editor

PLOS Genetics

Gregory P. Copenhaver

Editor-in-Chief

PLOS Genetics

Comments from the reviewers (if applicable):

The manuscript is now suitable for publication.

**Data Deposition**

http://datadryad.org/submit?journalID=pgenetics&manu=PGENETICS-D-21-00367R2

**Press Queries**

---

## [Editor Report · Acceptance letter]

3 Aug 2021

PGENETICS-D-21-00367R2 

Suppression of Vps13 adaptor protein mutants reveals a central role for PI4P in regulating prospore membrane extension 

Dear Dr TACHIKAWA, 

We are pleased to inform you that your manuscript entitled "Suppression of Vps13 adaptor protein mutants reveals a central role for PI4P in regulating prospore membrane extension" has been formally accepted for publication in PLOS Genetics! Your manuscript is now with our production department and you will be notified of the publication date in due course.

With kind regards,

Olena Szabo

PLOS Genetics

On behalf of:
